# Pontryagin Differentiable Programming:
# An End-to-End Learning and Control Framework

**Wanxin Jin**
Purdue University
{wanxinjin,zhaoranwang}@gmail.com

**Zhaoran Wang**
Northwestern University

**Zhuoran Yang**
Princeton University
zy6@princeton.edu

**Shaoshuai Mou**
Purdue University
mous@purdue.edu

## Abstract

This paper develops a Pontryagin Differentiable Programming (PDP) methodology, which establishes a unified framework to solve a broad class of learning and control tasks. The PDP distinguishes from existing methods by two novel techniques: first, we differentiate through Pontryagin's Maximum Principle, and this allows to obtain the analytical derivative of a trajectory with respect to tunable parameters within an optimal control system, enabling end-to-end learning of dynamics, policies, or/and control objective functions; and second, we propose an auxiliary control system in the backward pass of the PDP framework, and the output of this auxiliary control system is the analytical derivative of the original system's trajectory with respect to the parameters, which can be iteratively solved using standard control tools. We investigate three learning modes of the PDP: inverse reinforcement learning, system identification, and control/planning. We demonstrate the capability of the PDP in each learning mode on different high-dimensional systems, including multi-link robot arm, 6-DoF maneuvering quadrotor, and 6-DoF rocket powered landing.

## 1 Introduction

Many learning tasks can find their counterpart problems in control fields. These tasks both seek to obtain unknown aspects of a decision-making system with different terminologies compared below.

Table 1: Topic connections between control and learning (details presented in Section 2)

| UNKNOWNS IN A SYSTEM | LEARNING METHODS | CONTROL METHODS |
|---|---|---|
| Dynamics $\boldsymbol{x}_{t+1}=\boldsymbol{f}_{\boldsymbol{\theta}}(\boldsymbol{x}_t,\boldsymbol{u}_t)$ | Markov decision processes | System identification |
| Policy $\boldsymbol{u}_t = \boldsymbol{\pi}_{\boldsymbol{\theta}}(t,\boldsymbol{x}_t)$ | Reinforcement learning (RL) | Optimal control (OC) |
| Control objective $J=\sum_t c_{\boldsymbol{\theta}}(\boldsymbol{x}_t,\boldsymbol{u}_t)$ | Inverse RL | Inverse OC |

With the above connections, learning and control fields have begun to explore the complementary benefits of each other: control theory may provide abundant models and structures that allow for efficient or certificated algorithms for high-dimensional tasks, while learning enables to obtain these models from data, which are otherwise not readily attainable via classic control tools. Examples that enjoy both benefits include model-based RL [1, 2], where dynamics models are used for sample efficiency; and Koopman-operator control [3, 4], where via learning, nonlinear systems are lifted to a linear observable space to facilitate control design. Inspired by those, this paper aims to exploit the advantage of integrating learning and control and develop a unified framework that enables to solve a wide range of learning and control tasks, e.g., the challenging problems in Fig. 1.

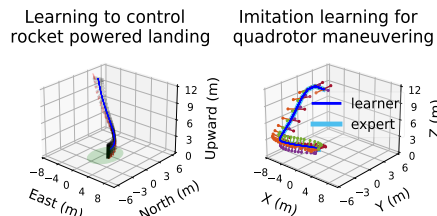

Learning to control rocket powered landing

Imitation learning for quadrotor maneuvering

Figure 1: left: PDP learns rocket landing control, right: PDP learns quadrotor dynamics and control objective for imitation.

## 2 Background and Related Work

**Learning dynamics.** This is usually referred as to as system identification in control fields, which typically consider linear systems represented by transfer functions [5]. For nonlinear systems, the Koopman theory [6] provides a way to lift states to a (infinite-dimensional) linear observable space [3, 7]. In learning, dynamics is characterized by Markov decision proceses and implemented using linear regression [8], observation-transition modeling [9], latent-space modeling [10], (deep) neural networks [11], Gaussian process [12], transition graphs [13], etc. Although off-the-shelf, most of these methods have to trade off between data efficiency and long-term prediction accuracy. To achieve both, physically-informed learning [14–17] injects physics laws into learning models, but they are limited to mechanical systems. Recently, a trend of work starts to use dynamical systems to explain (deep) neural networks, and some new algorithms [18–25] have been established.

This paper focuses on learning general dynamical models, encompassing either physical dynamics with unknown parameters or neural difference equations. The proposed learning framework is injected with inductive knowledge of optimal control theory to achieve efficiency and explainability.

**Learning optimal polices.** In learning fields, it relates to reinforcement learning (RL). Model-free RL provides a general-purpose framework to learn policies directly from interacting with environments [26–28], but usually suffers from significant data complexity. Model-based RL addresses this by first learning a dynamics model from experience and then integrating it to policy improvement [1, 12, 29–31]. The use of a model can assist to augment experience data [32, 33], perform back-propagation through time [12], or test policies before deployment. Model-based RL also faces some challenges that are not well-addressed. For example, how to efficiently leverage imperfect models [34], and how to maximize the joint benefit by combining policy learning and motion planning (trajectory optimization) [31, 35], where a policy has the advantage of execution coherence and fast deployment while the trajectory planning has the competence of adaption to unseen or future situations.

The counterpart topic in control is optimal control (OC), which is more concerned with characterizing optimal trajectories in presence of dynamics models. As in RL, the main strategy for OC is based on dynamic programming, and many valued-based methods are available, such as HJB [36], differential dynamical programming (DDP) [37] (by quadratizing dynamics and value function), and iterative linear quadratic regulator (iLQR) [38] (by linearizing dynamics and quadratizing value function). The second strategy to solve OC is based on the Pontryagin's Maximum/Minimal Principle (PMP) [39]. Derived from calculus of variations, PMP can be thought of as optimizing directly over trajectories, thus avoiding solving for value functions. Popular methods in this vein include shooting methods [40] and collocation methods [41]. However, the OC methods based on PMP are essentially *open loop* control and thus susceptible to model errors or disturbances in deployment. To address these, model predictive control (MPC) [42] generates controls given the system current state by repeatedly solving an OC problem over a finite prediction horizon (only the first optimal input is executed), leading to a *closed-loop* control form. Although MPC has dominated across many industrial applications [43], developing fast MPC implementations is still an active research direction [44].

The proposed learning framework in this work has a special mode for model-based control tasks. The method can be viewed as a complement to classic open-loop OC methods, because, although derived from PMP (trajectory optimization), the method here is to learn a *feedback/closed-loop* control policy. Depending on the specific policy parameterization, the method here can also be used for motion planning. All these features will provide new perspectives for model-based RL or MPC control.

**Learning control objective functions.** In learning, this relates to inverse reinforcement learning (IRL), whose goal is to find a control objective function to explain the given optimal demonstrations. The unknown objective function is typically parameterized as a weighted sum of features [45–47]. Strategies to learn the unknown weights include feature matching [45] (matching the feature values between demonstrations and reproduced trajectories), maximum entropy [46] (finding a trajectory distribution of maximum entropy subject to empirical feature values), and maximum margin [47] (maximizing the margin of objective values between demonstrations and reproduced trajectories). The learning update in the above IRL methods is preformed on a selected feature space by taking advantage of linearity of feature weights, and thus cannot be directly applied to learning objective functions that are nonlinear in parameters. The counterpart topic in the control field is inverse optimal control (IOC) [48–51]. With knowledge of dynamics, IOC focuses on more efficient learning paradigms. For example, by directly minimizing the violation of optimality conditions by the observed demonstration data, [48, 50–52] directly compute feature weights without repetitively solving the OC problems.

Despite the efficiency, minimizing optimality violation does not directly assure the closeness between the final reproduced trajectory and demonstrations or the closeness of their objective values.

Fundamentally different from existing IRL/IOC methods, this paper will develop a new framework that enables to learn complex control objective functions, e.g., neural objective functions, by directly minimizing the loss (e.g., the distance) between the reproduced trajectory and demonstrations.

**A unified perspective on learning dynamics/policy/control objective functions.** Consider a general decision-making system, which typically consists of aspects of dynamics, control policy, and control objective function. In a unified perspective, learning dynamics, policies, or control objective functions can be viewed as *instantiations of the same learning problem* but with (i) unknown parameters appearing in the system's different aspects and (ii) the different losses. For example, in learning dynamics, a differential/difference equation is parameterized and the loss function can be defined as the prediction error between the equation's output and target data; in learning policies, the unknown parameters are in a feedback policy and the loss function is just the control objective function; and in learning control objective functions, the control objective function is parameterized and the loss function can be the discrepancy between the reproduced trajectory and the observed demonstrations.

**Claim of contributions.** Motivated by the above, this paper develops a unified learning framework, named as PDP, that is flexible enough to be customized for different learning and control tasks and capable enough to efficiently solve high-dimensional and continuous-space problems. The proposed PDP framework borrows the idea of 'end-to-end' learning [53] and chooses to optimize a loss function directly with respect to the tunable parameters in the aspect(s) of a decision-making system, such as the dynamics, policy, or/and control objective function. The key contribution of the PDP is that we inject the optimal control theory as an inductive bias into the learning process to expedite the learning efficiency and explainability. Specifically, the PDP framework centers around the system's trajectory and *differentiates through PMP*, and this allows us to obtain the analytical derivative of the trajectory with respect to the tunable parameters, a key quantity for end-to-end learning of (neural) dynamics, (neural) policies, and (neural) control objective functions. Furthermore, we introduce an *auxiliary control system* in the back pass of the PDP framework, and its output trajectory is exactly the derivative of the trajectory with respect to the parameters, which can be iteratively solved using standard control tools. In control fields, to our best knowledge, this is the first work to propose the technique of the *differential PMP*, and more importantly, we show that the *differential PMP* can be easily obtained using the introduced auxiliary control system.

## 3 Problem Formulation

We begin with formulating a base problem and then discuss how to accommodate the base problem to specific applications. Consider a class of optimal control systems $\Sigma(\boldsymbol{\theta})$, which is parameterized by a tunable $\boldsymbol{\theta} \in \mathbb{R}^r$ in both dynamics and control (cost) objective function:

$$\Sigma(\boldsymbol{\theta}): \quad \begin{aligned} \text{dynamics:} \quad & \boldsymbol{x}_{t+1} = \boldsymbol{f}(\boldsymbol{x}_t, \boldsymbol{u}_t, \boldsymbol{\theta}) \quad \text{with given } \boldsymbol{x}_0, \\ \text{control objective:} \quad & J(\boldsymbol{\theta}) = \sum\nolimits_{t=0}^{T-1} c_t(\boldsymbol{x}_t, \boldsymbol{u}_t, \boldsymbol{\theta}) + h(\boldsymbol{x}_T, \boldsymbol{\theta}). \end{aligned} \quad (1)$$

Here, $\boldsymbol{x}_t \in \mathbb{R}^n$ is the system state; $\boldsymbol{u}_t \in \mathbb{R}^m$ is the control input; $\boldsymbol{f} : \mathbb{R}^n \times \mathbb{R}^m \times \mathbb{R}^r \mapsto \mathbb{R}^n$ is the dynamics model, which is assumed to be twice-differentiable; $t = 0, 1, \cdots, T$ is the time step with $T$ being the time horizon; and $J(\boldsymbol{\theta})$ is the control objective function with $c_t : \mathbb{R}^n \times \mathbb{R}^m \times \mathbb{R}^r \mapsto \mathbb{R}$ and $h : \mathbb{R}^n \times \mathbb{R}^r \mapsto \mathbb{R}$ denoting the stage/running and final costs, respectively, both of which are twice-differentiable. For a choice of $\boldsymbol{\theta}$, $\Sigma(\boldsymbol{\theta})$ will produce a trajectory of state-inputs:

$$\boldsymbol{\xi_\theta} = \{\boldsymbol{x}_{0:T}^{\boldsymbol{\theta}}, \boldsymbol{u}_{0:T-1}^{\boldsymbol{\theta}}\} \in \arg \min_{\{\boldsymbol{x}_{0:T}, \boldsymbol{u}_{0:T-1}\}} J(\boldsymbol{\theta}) \\ \text{subject to} \quad \boldsymbol{x}_{t+1} = \boldsymbol{f}(\boldsymbol{x}_t, \boldsymbol{u}_t, \boldsymbol{\theta}) \text{ for all } t \text{ given } \boldsymbol{x}_0 \quad , \quad (2)$$

that is, $\boldsymbol{\xi_\theta}$ optimizes $J(\boldsymbol{\theta})$ subject to the dynamics constraint $\boldsymbol{f}(\boldsymbol{\theta})$. For many applications (we will show next), one evaluates the above $\boldsymbol{\xi_\theta}$ using a scalar-valued differentiable loss $L(\boldsymbol{\xi_\theta}, \boldsymbol{\theta})$. Then, the **problem of interest** is to tune the parameter $\boldsymbol{\theta}$, such that $\boldsymbol{\xi_\theta}$ has the minimal loss:

$$\min_{\boldsymbol{\theta}} L(\boldsymbol{\xi_\theta}, \boldsymbol{\theta}) \quad \text{subject to} \quad \boldsymbol{\xi_\theta} \text{ is in } (2). \quad (3)$$

Under the above base formulation, for a specific learning or control task, one only needs to accordingly change precise details of $\Sigma(\boldsymbol{\theta})$ and define a specific loss function $L(\boldsymbol{\xi_\theta}, \boldsymbol{\theta})$, as we discuss below.

**IRL/IOC Mode.** Suppose that we are given optimal demonstrations $\boldsymbol{\xi}^{\mathrm{d}} = \{\boldsymbol{x}_{0:T}^{\mathrm{d}}, \boldsymbol{u}_{0:T-1}^{\mathrm{d}}\}$ of an expert optimal control system. We seek to learn the expert's dynamics and control objective function from $\boldsymbol{\xi}^{\mathrm{d}}$. To this end, we use $\boldsymbol{\Sigma}(\boldsymbol{\theta})$ in (1) to represent the expert, and define the loss in (3) as

$$L(\boldsymbol{\xi}_{\boldsymbol{\theta}}, \boldsymbol{\theta}) = l(\boldsymbol{\xi}_{\boldsymbol{\theta}}, \boldsymbol{\xi}^{\mathrm{d}}), \qquad (4)$$

where $l$ is a scalar function that penalizes the inconsistency of $\boldsymbol{\xi}_{\boldsymbol{\theta}}$ with $\boldsymbol{\xi}^{\mathrm{d}}$, e.g., $l(\boldsymbol{\xi}_{\boldsymbol{\theta}}, \boldsymbol{\xi}^{\mathrm{d}}) = \|\boldsymbol{\xi}_{\boldsymbol{\theta}} - \boldsymbol{\xi}^{\mathrm{d}}\|^2$. By solving (3) with (4), we can obtain a $\boldsymbol{\Sigma}(\boldsymbol{\theta}^*)$ whose trajectory is consistent with the observed demonstrations. It should be noted that even if the demonstrations $\boldsymbol{\xi}^{\mathrm{d}}$ significantly deviate from the optimal ones, the above formulation still finds the 'best' control objective function (and dynamics) within the parameterized set $\boldsymbol{\Sigma}(\boldsymbol{\theta})$ such that its reproduced $\boldsymbol{\xi}_{\boldsymbol{\theta}}$ in (2) has the *minimal distance* to $\boldsymbol{\xi}^{\mathrm{d}}$.

**SysID Mode.** Suppose that we are given data $\boldsymbol{\xi}^{\mathrm{o}} = \{\boldsymbol{x}_{0:T}^{\mathrm{o}}, \boldsymbol{u}_{0:T-1}\}$ collected from, say, a physical system (here, unlike $\boldsymbol{\xi}^{\mathrm{d}}$, $\boldsymbol{\xi}^{\mathrm{o}}$ is not necessarily optimal), and we wish to identify the system's dynamics. Here, $\boldsymbol{u}_{0:T-1}$ are usually externally supplied to ensure the physical system is of persistent excitation [54]. In order for $\boldsymbol{\Sigma}(\boldsymbol{\theta})$ in (1) to only represent dynamics (as we do not care about its internal control law), we set $J(\boldsymbol{\theta}) = 0$. Then, $\boldsymbol{\xi}_{\boldsymbol{\theta}}$ in (2) accepts any $\boldsymbol{u}_{0:T-1}^{\boldsymbol{\theta}} = \boldsymbol{u}_{0:T-1}$ as it always optimizes $J(\boldsymbol{\theta})=0$. In other words, by setting $J(\boldsymbol{\theta}) = 0$, $\boldsymbol{\Sigma}(\boldsymbol{\theta})$ in (1) now only represent a class of dynamics models:

$$\boldsymbol{\Sigma}(\boldsymbol{\theta}): \qquad \text{dynamics:} \quad \boldsymbol{x}_{t+1} = \boldsymbol{f}(\boldsymbol{x}_t, \boldsymbol{u}_t, \boldsymbol{\theta}) \qquad \text{with } \boldsymbol{x}_0 \text{ and } \boldsymbol{u}_{0:T-1}^{\boldsymbol{\theta}} = \boldsymbol{u}_{0:T-1}. \qquad (5)$$

Now, $\boldsymbol{\Sigma}(\boldsymbol{\theta})$ produces $\boldsymbol{\xi}_{\boldsymbol{\theta}} = \{\boldsymbol{x}_{0:T}^{\boldsymbol{\theta}}, \boldsymbol{u}_{0:T-1}^{\boldsymbol{\theta}}\}$ subject to (5). To use (3) for identifying $\boldsymbol{\theta}$, we define

$$L(\boldsymbol{\xi}_{\boldsymbol{\theta}}, \boldsymbol{\theta}) = l(\boldsymbol{\xi}_{\boldsymbol{\theta}}, \boldsymbol{\xi}^{\mathrm{o}}), \qquad (6)$$

where $l$ is to quantify the prediction error between $\boldsymbol{\xi}^{\mathrm{o}}$ and $\boldsymbol{\xi}_{\boldsymbol{\theta}}$ under the same inputs $\boldsymbol{u}_{0:T-1}$.

**Control/Planning Mode.** Consider a system with its dynamics learned in the above SysID. We want to obtain a *feedback controller* or *trajectory* such that the system achieves a performance of minimizing a given cost function. To that end, we specialize $\boldsymbol{\Sigma}(\boldsymbol{\theta})$ in (1) as follows: first, set $\boldsymbol{f}$ as the learned dynamics and $J(\boldsymbol{\theta}) = 0$; and second, through a *close-loop link*, we connect the input $\boldsymbol{u}_t$ and state $\boldsymbol{x}_t$ via a parameterized policy block $\boldsymbol{u}_t = \boldsymbol{u}(t, \boldsymbol{x}_t, \boldsymbol{\theta})$ (reminder: unlike SysID Mode with $\boldsymbol{u}_t$ supplied externally, the inputs here are from a policy via a feedback loop). $\boldsymbol{\Sigma}(\boldsymbol{\theta})$ now becomes

$$\boldsymbol{\Sigma}(\boldsymbol{\theta}): \qquad \begin{array}{ll} \text{dynamics:} & \boldsymbol{x}_{t+1} = \boldsymbol{f}(\boldsymbol{x}_t, \boldsymbol{u}_t) \quad \text{with} \quad \boldsymbol{x}_0, \\ \text{control policy:} & \boldsymbol{u}_t = \boldsymbol{u}(t, \boldsymbol{x}_t, \boldsymbol{\theta}). \end{array} \qquad (7)$$

Now, $\boldsymbol{\Sigma}(\boldsymbol{\theta})$ produces a trajectory $\boldsymbol{\xi}_{\boldsymbol{\theta}} = \{\boldsymbol{x}_{0:T}^{\boldsymbol{\theta}}, \boldsymbol{u}_{0:T-1}^{\boldsymbol{\theta}}\}$ subject to (7). We set the loss in (3) as

$$L(\boldsymbol{\xi}_{\boldsymbol{\theta}}, \boldsymbol{\theta}) = \sum_{t=0}^{T-1} l(\boldsymbol{x}_t^{\boldsymbol{\theta}}, \boldsymbol{u}_t^{\boldsymbol{\theta}}) + l_f(\boldsymbol{x}_T^{\boldsymbol{\theta}}), \qquad (8)$$

where $l$ and $l_f$ are the stage and final costs, respectively. Then, (3) is an optimal control or planning problem: if $\boldsymbol{u}_t = \boldsymbol{u}(t, \boldsymbol{x}_t, \boldsymbol{\theta})$ (i.e., feedback policy explicitly depends on $\boldsymbol{x}_t$), (3) is a *close-loop optimal control* problem; otherwise if $\boldsymbol{u}_t = \boldsymbol{u}(t, \boldsymbol{\theta})$ (e.g., polynomial parameterization), (3) is an *open-loop motion planning* problem. This mode can also be used as a component to solve (1) in IRL/IOC Mode.

## 4  An End-to-End Learning Framework

To solve the generic problem in (3), the idea of end-to-end learning [53] seeks to optimize the loss $L(\boldsymbol{\xi}_{\boldsymbol{\theta}}, \boldsymbol{\theta})$ *directly* with respect to the tunable parameter $\boldsymbol{\theta}$, by applying the gradient descent

$$\boldsymbol{\theta}_{k+1} = \boldsymbol{\theta}_k - \eta_k \frac{dL}{d\boldsymbol{\theta}}\Big|_{\boldsymbol{\theta}_k} \quad \text{with} \quad \frac{dL}{d\boldsymbol{\theta}}\Big|_{\boldsymbol{\theta}_k} = \frac{\partial L}{\partial \boldsymbol{\xi}}\Big|_{\boldsymbol{\xi}_{\boldsymbol{\theta}_k}} \frac{\partial \boldsymbol{\xi}_{\boldsymbol{\theta}}}{\partial \boldsymbol{\theta}}\Big|_{\boldsymbol{\theta}_k} + \frac{\partial L}{\partial \boldsymbol{\theta}}\Big|_{\boldsymbol{\theta}_k}. \qquad (9)$$

Here, $k = 0, 1, \cdots$ is the iteration index; $\frac{dL}{d\boldsymbol{\theta}}\big|_{\boldsymbol{\theta}_k}$ is the gradient of the loss with respect to $\boldsymbol{\theta}$ evaluated at $\boldsymbol{\theta}_k$; and $\eta_k$ is the learning rate. From (9), we can draw a learning architecture in Fig. 2. Each update of $\boldsymbol{\theta}$ consists of a *forward pass*, where at $\boldsymbol{\theta}_k$, the corresponding trajectory $\boldsymbol{\xi}_{\boldsymbol{\theta}_k}$ is solved from $\boldsymbol{\Sigma}(\boldsymbol{\theta}_k)$ and the loss is computed, and a *backward pass*, where $\frac{\partial L}{\partial \boldsymbol{\xi}}\big|_{\boldsymbol{\xi}_{\boldsymbol{\theta}_k}}$, $\frac{\partial \boldsymbol{\xi}_{\boldsymbol{\theta}}}{\partial \boldsymbol{\theta}}\big|_{\boldsymbol{\theta}_k}$, and $\frac{\partial L}{\partial \boldsymbol{\theta}}\big|_{\boldsymbol{\theta}_k}$ are computed.

In the forward pass, $\boldsymbol{\xi}_{\boldsymbol{\theta}}$ is obtained by solving an optimal control problem in $\boldsymbol{\Sigma}(\boldsymbol{\theta})$ using any available OC methods, such as iLQR or Control/Planning Mode, (note that in SysID or Control/Planning modes, it is reduced to integrating difference equations (5) or (7)). In backward pass, $\frac{\partial L}{\partial \boldsymbol{\xi}}$ and $\frac{\partial L}{\partial \boldsymbol{\theta}}$ are easily obtained from the loss function $L(\boldsymbol{\xi}_{\boldsymbol{\theta}}, \boldsymbol{\theta})$. The main challenge, however, is to solve $\frac{\partial \boldsymbol{\xi}_{\boldsymbol{\theta}}}{\partial \boldsymbol{\theta}}$, i.e., *the derivative of a trajectory with respect to the parameters in the system*. Next, we will analytically solve $\frac{\partial \boldsymbol{\xi}_{\boldsymbol{\theta}}}{\partial \boldsymbol{\theta}}$ by proposing two techniques: *differential PMP* and *auxiliary control system*.

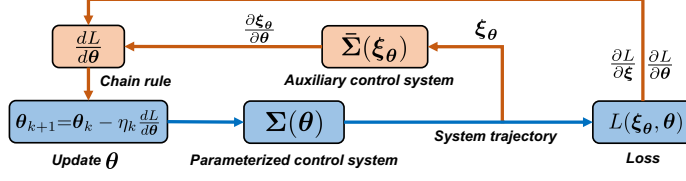

Figure 2: PDP end-to-end learning framework.

# 5   Key Contributions: Differential PMP & Auxiliary Control System

We first recall the discrete-time Pontryagin's Maximum/Minimum Principle (PMP) [39] (a derivation of discrete-time PMP is given in Appendix C). For the optimal control system $\boldsymbol{\Sigma}(\boldsymbol{\theta})$ in (1) with a fixed $\boldsymbol{\theta}$, PMP describes a set of optimality conditions which the trajectory $\boldsymbol{\xi_\theta} = \{\boldsymbol{x}_{0:T}^{\boldsymbol{\theta}}, \boldsymbol{u}_{0:T-1}^{\boldsymbol{\theta}}\}$ in (2) must satisfy. To introduce these conditions, we first define the following *Hamiltonian*,

$$H_t = c_t(\boldsymbol{x}_t, \boldsymbol{u}_t; \boldsymbol{\theta}) + \boldsymbol{f}(\boldsymbol{x}_t, \boldsymbol{u}_t; \boldsymbol{\theta})' \boldsymbol{\lambda}_{t+1}, \tag{10}$$

where $\boldsymbol{\lambda}_t \in \mathbb{R}^n$ ($t = 1, 2, \cdots, T$) is called the *costate variable*, which can be also thought of as the Lagrange multipliers for the dynamics constraints. According to PMP, there exists a sequence of costates $\boldsymbol{\lambda}_{1:T}^{\boldsymbol{\theta}}$, which together with the optimal trajectory $\boldsymbol{\xi_\theta} = \{\boldsymbol{x}_{0:T}^{\boldsymbol{\theta}}, \boldsymbol{u}_{0:T-1}^{\boldsymbol{\theta}}\}$ satisfy

$$\text{dynamics equation:} \qquad \boldsymbol{x}_{t+1}^{\boldsymbol{\theta}} = \frac{\partial H_t}{\partial \boldsymbol{\lambda}_{t+1}^{\boldsymbol{\theta}}} = \boldsymbol{f}(\boldsymbol{x}_t^{\boldsymbol{\theta}}, \boldsymbol{u}_t^{\boldsymbol{\theta}}; \boldsymbol{\theta}), \tag{11a}$$

$$\text{costate equation:} \qquad \boldsymbol{\lambda}_t^{\boldsymbol{\theta}} = \frac{\partial H_t}{\partial \boldsymbol{x}_t^{\boldsymbol{\theta}}} = \frac{\partial c_t}{\partial \boldsymbol{x}_t^{\boldsymbol{\theta}}} + \frac{\partial \boldsymbol{f}'}{\partial \boldsymbol{x}_t^{\boldsymbol{\theta}}} \boldsymbol{\lambda}_{t+1}^{\boldsymbol{\theta}}, \tag{11b}$$

$$\text{input equation:} \qquad \boldsymbol{0} = \frac{\partial H_t}{\partial \boldsymbol{u}_t^{\boldsymbol{\theta}}} = \frac{\partial c_t}{\partial \boldsymbol{u}_t^{\boldsymbol{\theta}}} + \frac{\partial \boldsymbol{f}'}{\partial \boldsymbol{u}_t^{\boldsymbol{\theta}}} \boldsymbol{\lambda}_{t+1}^{\boldsymbol{\theta}}, \tag{11c}$$

$$\text{boundary conditions:} \qquad \boldsymbol{\lambda}_T^{\boldsymbol{\theta}} = \frac{\partial h}{\partial \boldsymbol{x}_T^{\boldsymbol{\theta}}}, \qquad \boldsymbol{x}_0^{\boldsymbol{\theta}} = \boldsymbol{x}_0. \tag{11d}$$

For notation simplicity, $\frac{\partial \boldsymbol{g}}{\partial \boldsymbol{x}_t}$ means the derivative of function $\boldsymbol{g}(\boldsymbol{x})$ with respect to $\boldsymbol{x}$ evaluated at $\boldsymbol{x}_t$.

## 5.1   Differential PMP

To begin, recall that our goal (in Section 4) is to obtain $\frac{\partial \boldsymbol{\xi_\theta}}{\partial \boldsymbol{\theta}}$, that is,

$$\frac{\partial \boldsymbol{\xi_\theta}}{\partial \boldsymbol{\theta}} = \left\{ \frac{\partial \boldsymbol{x}_{0:T}^{\boldsymbol{\theta}}}{\partial \boldsymbol{\theta}}, \frac{\partial \boldsymbol{u}_{0:T-1}^{\boldsymbol{\theta}}}{\partial \boldsymbol{\theta}} \right\}. \tag{12}$$

To this end, we are motivated to differentiate the PMP conditions in (11) on both sides with respect to $\boldsymbol{\theta}$. This leads to the following *differential PMP*:

$$\text{differential dynamics equation:} \qquad \frac{\partial \boldsymbol{x}_{t+1}^{\boldsymbol{\theta}}}{\partial \boldsymbol{\theta}} = F_t \frac{\partial \boldsymbol{x}_t^{\boldsymbol{\theta}}}{\partial \boldsymbol{\theta}} + G_t \frac{\partial \boldsymbol{u}_t^{\boldsymbol{\theta}}}{\partial \boldsymbol{\theta}} + E_t, \tag{13a}$$

$$\text{differential costate equation:} \qquad \frac{\partial \boldsymbol{\lambda}_t^{\boldsymbol{\theta}}}{\partial \boldsymbol{\theta}} = H_t^{xx} \frac{\partial \boldsymbol{x}_t^{\boldsymbol{\theta}}}{\partial \boldsymbol{\theta}} + H_t^{xu} \frac{\partial \boldsymbol{u}_t^{\boldsymbol{\theta}}}{\partial \boldsymbol{\theta}} + F_t' \frac{\partial \boldsymbol{\lambda}_{t+1}^{\boldsymbol{\theta}}}{\partial \boldsymbol{\theta}} + H_t^{xe}, \tag{13b}$$

$$\text{differential input equation:} \qquad \boldsymbol{0} = H_t^{ux} \frac{\partial \boldsymbol{x}_t^{\boldsymbol{\theta}}}{\partial \boldsymbol{\theta}} + H_t^{uu} \frac{\partial \boldsymbol{u}_t^{\boldsymbol{\theta}}}{\partial \boldsymbol{\theta}} + G_t' \frac{\partial \boldsymbol{\lambda}_{t+1}^{\boldsymbol{\theta}}}{\partial \boldsymbol{\theta}} + H_t^{ue}, \tag{13c}$$

$$\text{differential boundary conditions:} \qquad \frac{\partial \boldsymbol{\lambda}_T^{\boldsymbol{\theta}}}{\partial \boldsymbol{\theta}} = H_T^{xx} \frac{\partial \boldsymbol{x}_T^{\boldsymbol{\theta}}}{\partial \boldsymbol{\theta}} + H_T^{xe}, \qquad \frac{\partial \boldsymbol{x}_0^{\boldsymbol{\theta}}}{\partial \boldsymbol{\theta}} = \frac{\partial \boldsymbol{x}_0}{\partial \boldsymbol{\theta}} = \boldsymbol{0}. \tag{13d}$$

Here, to simplify notations and distinguish knowns and unknowns, the coefficient matrices in the above differential PMP (13) are defined as follows:

$$F_t = \frac{\partial \boldsymbol{f}}{\partial \boldsymbol{x}_t^{\boldsymbol{\theta}}}, \qquad G_t = \frac{\partial \boldsymbol{f}}{\partial \boldsymbol{u}_t^{\boldsymbol{\theta}}}, \qquad H_t^{xx} = \frac{\partial^2 H_t}{\partial \boldsymbol{x}_t^{\boldsymbol{\theta}} \partial \boldsymbol{x}_t^{\boldsymbol{\theta}}}, \qquad H_t^{xe} = \frac{\partial^2 H_t}{\partial \boldsymbol{x}_t^{\boldsymbol{\theta}} \partial \boldsymbol{\theta}}, \qquad H_t^{xu} = \frac{\partial^2 H_t}{\partial \boldsymbol{x}_t^{\boldsymbol{\theta}} \partial \boldsymbol{u}_t^{\boldsymbol{\theta}}} = (H_t^{ux})', \tag{14a}$$

$$E_t = \frac{\partial \boldsymbol{f}}{\partial \boldsymbol{\theta}}, \qquad H_t^{uu} = \frac{\partial^2 H_t}{\partial \boldsymbol{u}_t^{\boldsymbol{\theta}} \partial \boldsymbol{u}_t^{\boldsymbol{\theta}}}, \qquad H_t^{ue} = \frac{\partial^2 H_t}{\partial \boldsymbol{u}_t^{\boldsymbol{\theta}} \partial \boldsymbol{\theta}}, \qquad H_T^{xx} = \frac{\partial^2 h}{\partial \boldsymbol{x}_T^{\boldsymbol{\theta}} \partial \boldsymbol{x}_T^{\boldsymbol{\theta}}}, \qquad H_T^{xe} = \frac{\partial^2 h}{\partial \boldsymbol{x}_T^{\boldsymbol{\theta}} \partial \boldsymbol{\theta}}, \tag{14b}$$

where we use $\frac{\partial^2 \boldsymbol{g}}{\partial \boldsymbol{x}_t \partial \boldsymbol{u}_t}$ to denote the second-order derivative of a function $\boldsymbol{g}(\boldsymbol{x}, \boldsymbol{u})$ evaluated at $(\boldsymbol{x}_t, \boldsymbol{u}_t)$. Since the trajectory $\boldsymbol{\xi_\theta} = \{\boldsymbol{x}_{0:T}^{\boldsymbol{\theta}}, \boldsymbol{u}_{0:T-1}^{\boldsymbol{\theta}}\}$ is obtained in the forward pass (recall Fig. 2), all matrices

in (14) are thus known (note that the computation of these matrices also requires $\boldsymbol{\lambda}_{1:T}^{\boldsymbol{\theta}}$, which can be obtained by iteratively solving (11b) and (11d) given $\boldsymbol{\xi_\theta}$). From the differential PMP in (13), we note that to obtain $\frac{\partial \boldsymbol{\xi_\theta}}{\partial \boldsymbol{\theta}}$ in (12), it is sufficient to compute the unknowns $\left\{ \frac{\partial \boldsymbol{x}_{0:T}^{\boldsymbol{\theta}}}{\partial \boldsymbol{\theta}}, \frac{\partial \boldsymbol{x}_{0:T-1}^{\boldsymbol{\theta}}}{\partial \boldsymbol{\theta}}, \frac{\partial \boldsymbol{\lambda}_{1:T}^{\boldsymbol{\theta}}}{\partial \boldsymbol{\theta}} \right\}$ in (13). Next we will show that how these unknowns are elegantly solved by introducing a new system.

## 5.2 Auxiliary Control System

One important observation to the differential PMP in (13) is that it shares a similar structure to the original PMP in (11); so it can be viewed as a new set of PMP equations corresponding to an 'oracle control optimal system' whose the 'optimal trajectory' is exactly (12). This motivates us to 'unearth' this oracle optimal control system, because by doing so, (12) can be obtained from this oracle system by an OC solver. To this end, we first define the new 'state' and 'control' (matrix) variables:

$$X_t = \frac{\partial \boldsymbol{x}_t}{\partial \boldsymbol{\theta}} \in \mathbb{R}^{n \times r}, \qquad U_t = \frac{\partial \boldsymbol{u}_t}{\partial \boldsymbol{\theta}} \in \mathbb{R}^{m \times r}, \tag{15}$$

respectively. Then, we 'artificially' define the following *auxiliary control system* $\overline{\boldsymbol{\Sigma}}(\boldsymbol{\xi_\theta})$:

$$\overline{\boldsymbol{\Sigma}}(\boldsymbol{\xi_\theta}):
\begin{array}{rl}
\text{dynamics:} & X_{t+1} = F_t X_t + G_t U_t + E_t \quad \text{with} \quad X_0 = \mathbf{0}, \\[2mm]
\text{control objective:} & \bar{J} = \mathrm{Tr} \sum_{t=0}^{T-1} \left( \frac{1}{2} \begin{bmatrix} X_t \\ U_t \end{bmatrix}' \begin{bmatrix} H_t^{xx} & H_t^{xu} \\ H_t^{ux} & H_t^{uu} \end{bmatrix} \begin{bmatrix} X_t \\ U_t \end{bmatrix} + \begin{bmatrix} H_t^{xe} \\ H_t^{ue} \end{bmatrix}' \begin{bmatrix} X_t \\ U_t \end{bmatrix} \right) \\[4mm]
& + \mathrm{Tr} \left( \frac{1}{2} X_T' H_T^{xx} U_T + (H_T^{xe})' X_T \right).
\end{array} \tag{16}$$

Here, $X_0 = \frac{\partial \boldsymbol{x}_0}{\partial \boldsymbol{\theta}} = \mathbf{0}$ because $\boldsymbol{x}_0$ in (1) is given; $\bar{J}$ is the defined control objective function which needs to be optimized in the auxiliary control system; and Tr denotes matrix trace. Before presenting the key results, we make some comments on the above auxiliary control system $\overline{\boldsymbol{\Sigma}}(\boldsymbol{\xi_\theta})$. First, its state and control variables are both matrix variables defined in (15). Second, its dynamics is linear and control objective function $\bar{J}$ is quadratic, for which the coefficient matrices are given in (14). Third, its dynamics and objective function are determined by the trajectory $\boldsymbol{\xi_\theta}$ of the system $\boldsymbol{\Sigma}(\boldsymbol{\theta})$ in forward pass, and this is why we denote it as $\overline{\boldsymbol{\Sigma}}(\boldsymbol{\xi_\theta})$. Finally, we have the following important result.

**Lemma 5.1.** *Let* $\{X_{0:T}^{\boldsymbol{\theta}}, U_{0:T-1}^{\boldsymbol{\theta}}\}$ *be a stationary solution to the auxiliary control system* $\overline{\boldsymbol{\Sigma}}(\boldsymbol{\xi_\theta})$ *in (16). Then,* $\{X_{0:T}^{\boldsymbol{\theta}}, U_{0:T-1}^{\boldsymbol{\theta}}\}$ *satisfies Pontryagin's Maximum Principle of* $\overline{\boldsymbol{\Sigma}}(\boldsymbol{\xi_\theta})$*, which is (13), and*

$$\{X_{0:T}^{\boldsymbol{\theta}}, U_{0:T-1}^{\boldsymbol{\theta}}\} = \left\{ \frac{\partial \boldsymbol{x}_{0:T}^{\boldsymbol{\theta}}}{\partial \boldsymbol{\theta}}, \frac{\partial \boldsymbol{u}_{0:T-1}^{\boldsymbol{\theta}}}{\partial \boldsymbol{\theta}} \right\} = \frac{\partial \boldsymbol{\xi_\theta}}{\partial \boldsymbol{\theta}}. \tag{17}$$

A proof of Lemma 5.1 is in Appendix A. Lemma 5.1 states two assertions. First, the PMP condition for the auxiliary control system $\overline{\boldsymbol{\Sigma}}(\boldsymbol{\xi_\theta})$ is exactly the differential PMP in (13) for the original system $\boldsymbol{\Sigma}(\boldsymbol{\theta})$; and second, importantly, the trajectory $\{X_{0:T}^{\boldsymbol{\theta}}, U_{0:T-1}^{\boldsymbol{\theta}}\}$ produced by the auxiliary control system $\overline{\boldsymbol{\Sigma}}(\boldsymbol{\xi_\theta})$ is exactly the derivative of trajectory of the original system $\boldsymbol{\Sigma}(\boldsymbol{\theta})$ with respect to the parameter $\boldsymbol{\theta}$. Based on Lemma 5.1, we can obtain $\frac{\partial \boldsymbol{\xi_\theta}}{\partial \boldsymbol{\theta}}$ from $\overline{\boldsymbol{\Sigma}}(\boldsymbol{\xi_\theta})$ efficiently by the lemma below.

**Lemma 5.2.** *If* $H_t^{uu}$ *in (16) is invertible for all* $t = 0, 1 \cdots, T-1$*, define the following recursions*

$$P_t = Q_t + A_t'(I + P_{t+1}R_t)^{-1} P_{t+1} A_t, \tag{18a}$$

$$W_t = A_t'(I + P_{t+1}R_t)^{-1}(W_{t+1} + P_{t+1}M_t) + N_t, \tag{18b}$$

*with* $P_T = H_T^{xx}$ *and* $W_T = H_T^{xe}$*. Here,* $I$ *is identity matrix,* $A_t = F_t - G_t(H_t^{uu})^{-1} H_t^{ux}, R_t = G_t(H_t^{uu})^{-1} G_t', M_t = E_t - G_t(H_t^{uu})^{-1} H_t^{ue}, Q_t = H_t^{xx} - H_t^{xu}(H_t^{uu})^{-1} H_t^{ux}, N_t = H_t^{xe} - H_t^{xu}(H_t^{uu})^{-1} H_t^{ue}$ *are all known given (14). Then, the stationary solution* $\{X_{0:T}^{\boldsymbol{\theta}}, U_{0:T-1}^{\boldsymbol{\theta}}\}$ *in (17) can be obtained by iteratively solving the following equations from* $t = 0$ *to* $T-1$ *with* $X_0^{\boldsymbol{\theta}} = X_0 = \mathbf{0}$*:*

$$U_t^{\boldsymbol{\theta}} = -(H_t^{uu})^{-1} \left( H_t^{ux} X_t^{\boldsymbol{\theta}} + H_t^{ue} + G_t'(I + P_{t+1}R_t)^{-1} \left( P_{t+1} A_t X_t^{\boldsymbol{\theta}} + P_{t+1} M_t + W_{t+1} \right) \right), \tag{19a}$$

$$X_{t+1}^{\boldsymbol{\theta}} = F_t X_t^{\boldsymbol{\theta}} + G_t U_t^{\boldsymbol{\theta}} + E_t. \tag{19b}$$

A proof of Lemma 5.2 is in Appendix B. Lemma 5.2 states that the trajectory of the above auxiliary control system $\overline{\boldsymbol{\Sigma}}(\boldsymbol{\xi_\theta})$ can be obtained by two steps: first, iteratively solve (18) backward in time to

obtain matrices $P_t$ and $W_t$ (all other coefficient matrices are known given $\overline{\Sigma}(\boldsymbol{\xi_\theta})$); second, calculate $\{X_{0:T}^{\boldsymbol{\theta}}, U_{0:T-1}^{\boldsymbol{\theta}}\}$ by iteratively integrating a feedback-control system (19) forward in time. In fact, these two steps constitute the standard procedure to solve general finite-time LQR problems [55].

As a conclusion to the techniques developed in Section 5, in Algorithm 1 we summarize the procedure of computing $\frac{\partial \boldsymbol{\xi_\theta}}{\partial \boldsymbol{\theta}}$ via the introduced auxiliary control system. Algorithm 1 serves as a key component in the backward pass of the PDP learning framework, as shown in Fig. 2.

---

**Algorithm 1:** Solving $\frac{\partial \boldsymbol{\xi_\theta}}{\partial \boldsymbol{\theta}}$ using Auxiliary Control System     (*See detailed version in Appendix D* )

---

**Input:** The trajectory $\boldsymbol{\xi_\theta}$ in (2) produced by the system $\Sigma(\boldsymbol{\theta})$ in (1) in the forward pass.

         Compute the coefficient matrices (14) to obtain the auxiliary control system $\overline{\Sigma}(\boldsymbol{\xi_\theta})$ in (16);

         Solve the auxiliary control system $\overline{\Sigma}(\boldsymbol{\xi_\theta})$ to obtain $\{X_{0:T}^{\boldsymbol{\theta}}, U_{0:T-1}^{\boldsymbol{\theta}}\}$ using Lemma 5.2;

**Return:** $\frac{\partial \boldsymbol{\xi_\theta}}{\partial \boldsymbol{\theta}} = \{X_{0:T}^{\boldsymbol{\theta}}, U_{0:T-1}^{\boldsymbol{\theta}}\}$

---

# 6 Applications to Different Learning Modes and Experiments

We investigate three learning modes of PDP, as described in Section 3. For each mode, we demonstrate its capability in four environments listed in Table 2, and a baseline and a state-of-the-art method are compared. Both PDP and environment codes are available at `https://github.com/wanxinjin`.

Table 2: Experimental environments (results for 6-DoF rocket landing is in Appendix I)

| Systems | Dynamics parameter $\boldsymbol{\theta}_{\text{dyn}}$ | Control objective parameter $\boldsymbol{\theta}_{\text{obj}}$ |
|---|---|---|
| Cartpole | cart mass, pole mass and length | |
| Two-link robot arm | length and mass for each link | $c(\boldsymbol{x}, \boldsymbol{u}) = \|\boldsymbol{\theta}_{\text{obj}}'(\boldsymbol{x} - \boldsymbol{x}_{\text{g}})\|^2 + \|\boldsymbol{u}\|^2$ |
| 6-DoF quadrotor maneuvering | mass, wing length, inertia matrix | $h(\boldsymbol{x}, \boldsymbol{u}) = \|\boldsymbol{\theta}_{\text{obj}}'(\boldsymbol{x} - \boldsymbol{x}_{\text{g}})\|^2$ |
| 6-DoF rocket powered landing | mass, rocket length, inertia matrix | |

We fix the unit weight to $\|\boldsymbol{u}\|^2$, because estimating all weights will incur ambiguity [48]; $\boldsymbol{x}_{\text{g}}$ is the goal state.

**IRL/IOC Mode.** The parameterized $\Sigma(\boldsymbol{\theta})$ is in (1) and the loss in (4). In the forward pass of PDP, $\boldsymbol{\xi_\theta}$ is solved from $\Sigma(\boldsymbol{\theta})$ by any OC solver. In the backward pass, $\frac{\partial \boldsymbol{\xi_\theta}}{\partial \boldsymbol{\theta}}$ is computed from the auxiliary control system $\overline{\Sigma}(\boldsymbol{\xi_\theta})$ in (16) using Algorithm 1. The full algorithm is in Appendix D.

**Experiment: imitation learning.** We use IRL/IOC Mode to solve imitation learning in environments in Table 2. The true dynamics is parameterized, and control objective is parameterized as a weighted distance to the goal, $\boldsymbol{\theta} = \{\boldsymbol{\theta}_{\text{dyn}}, \boldsymbol{\theta}_{\text{obj}}\}$. Set imitation loss $L(\boldsymbol{\xi_\theta}, \boldsymbol{\theta}) = \|\boldsymbol{\xi}^{\text{d}} - \boldsymbol{\xi_\theta}\|^2$. Two other methods are compared: (i) neural policy cloning, and (ii) inverse KKT [52]. We set learning rate $\eta = 10^{-4}$ and run five trials given random initial $\boldsymbol{\theta}_0$. The results in Fig. 3a-3c show that PDP significantly outperforms the policy cloning and inverse-KKT for a much lower training loss and faster convergence. In Fig. 3d, we apply the PDP to learn a neural control objective function for the robot arm using the same demonstration data in Fig. 3b, and we also compare with the GAIL [56]. Results in Fig. 3d show that the PDP successfully learns a neural objective function and the imitation loss of PDP is much lower than that of GAIL. It should note that because the demonstrations are not strictly realizable (optimal) under the parameterized neural objective function, the final loss for the PDP is small but not zero. This indicates that given sub-optimal demonstrations, PDP can still find the 'best' control objective function within the function set $J(\boldsymbol{\theta})$ such that its reproduced $\boldsymbol{\xi_\theta}$ has the *minimal distance* to the demonstrations. Please refer to Appendix E.2 for more experiment details and additional validations.

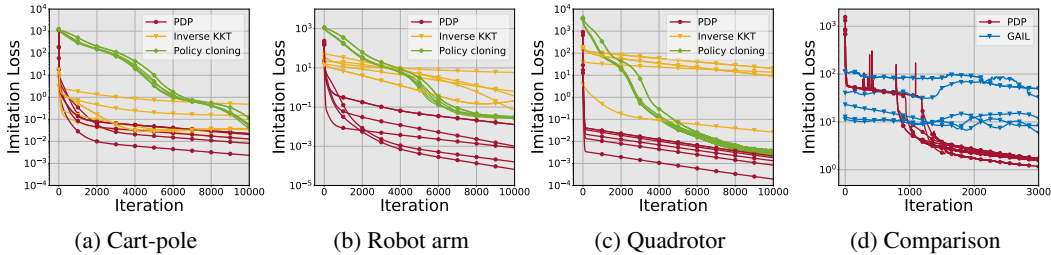

| (a) Cart-pole | (b) Robot arm | (c) Quadrotor | (d) Comparison |

Figure 3: (a-c) imitation loss v.s. iteration, (d) PDP learns a neural objective function and comparison.

**SysID Mode.** In this mode, $\Sigma(\boldsymbol{\theta})$ is (5) and loss is (6). PDP is greatly simplified: in forward pass, $\boldsymbol{\xi_\theta}$ is solved by integrating the difference equation (5). In the backward pass, $\overline{\Sigma}(\boldsymbol{\xi_\theta})$ is reduced to

$$\overline{\Sigma}(\boldsymbol{\xi_\theta}): \qquad\qquad \text{dynamics:} \quad X_{t+1}^{\boldsymbol{\theta}} = F_t X_t^{\boldsymbol{\theta}} + E_t \quad \text{with} \quad X_0 = \mathbf{0}. \qquad\qquad (20)$$

This is because $\boldsymbol{\Sigma}(\boldsymbol{\theta})$ in (5) results from letting $J(\boldsymbol{\theta})=0$, (13b-13d) and $\bar{J}$ in (16) are then trivialized, and due to $\boldsymbol{u}_{0:T-1}$ given, $U_t^{\boldsymbol{\theta}}=\mathbf{0}$ in (13a). The algorithm is in Appendix D.

**Experiment: system identification.** We use the SysID Mode to identify the dynamics parameter $\boldsymbol{\theta}_{\text{dyn}}$ for the systems in Table 2. Set the SysID loss $L(\boldsymbol{\xi}_{\boldsymbol{\theta}}, \boldsymbol{\theta}) = \|\boldsymbol{\xi}^{\text{o}} - \boldsymbol{\xi}_{\boldsymbol{\theta}}\|^2$. Two other methods are compared: (i) learning a neural network (NN) dynamics model, and (ii) DMDc [57]. For all methods, we set learning rate $\eta = 10^{-4}$, and run five trials with random $\boldsymbol{\theta}_0$. The results are in Fig. 4. Fig. 4a-4c show an obvious advantage of PDP over the NN baseline and DMDc in terms of lower training loss and faster convergence speed. In Fig. 4d, we compare PDP and Adam [58] (here both with $\eta = 10^{-5}$) for training the same neural dynamics model for the robot arm. The results again show that PDP outperforms Adam for faster learning speed and lower training loss. Such advantages are due to that PDP has injected an inductive bias of optimal control into learning, making it more efficient for handling dynamical systems. More experiments and validations are in Appendix E.3.

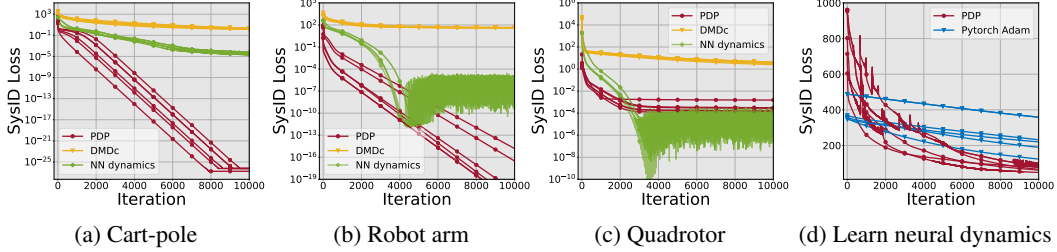

(a) Cart-pole      (b) Robot arm      (c) Quadrotor      (d) Learn neural dynamics

Figure 4: (a-c) SysID loss v.s. iteration, (d) PDP learns a neural dynamics model.

**Control/Planning Mode.** The parameterized system $\boldsymbol{\Sigma}(\boldsymbol{\theta})$ is (7) and loss is (8). PDP for this mode is also simplified. In forward pass, $\boldsymbol{\xi}_{\boldsymbol{\theta}}$ is solved by integrating a (controlled) difference equation (7). In backward pass, $\bar{J}$ in the auxiliary control system (16) is trivialized because we have considered $J(\boldsymbol{\theta}) = 0$ in (7). Since the control is now given by $\boldsymbol{u}_t = \boldsymbol{u}(t, \boldsymbol{x}_t, \boldsymbol{\theta})$, $U_t^{\boldsymbol{\theta}}$ is obtained by differentiating the policy on both side with respect to $\boldsymbol{\theta}$, that is, $U_t^{\boldsymbol{\theta}} = U_t^x X_t^{\boldsymbol{\theta}} + U_t^e$ with $U_t^x = \frac{\partial \boldsymbol{u}_t}{\partial \boldsymbol{x}_t}$ and $U_t^e = \frac{\partial \boldsymbol{u}_t}{\partial \boldsymbol{\theta}}$. Thus,

$$\overline{\boldsymbol{\Sigma}}(\boldsymbol{\xi}_{\boldsymbol{\theta}}): \quad \begin{array}{ll} \text{dynamics:} & X_{t+1}^{\boldsymbol{\theta}} = F_t X_t^{\boldsymbol{\theta}} + G_t U_t^{\boldsymbol{\theta}} \quad \text{with} \quad X_0 = \mathbf{0}, \\ \text{control policy:} & U_t^{\boldsymbol{\theta}} = U_t^x X_t^{\boldsymbol{\theta}} + U_t^e. \end{array} \tag{21}$$

Integrating (21) from $t = 0$ to $T$ leads to $\{X_{0:T}^{\boldsymbol{\theta}}, U_{0:T-1}^{\boldsymbol{\theta}}\} = \frac{\partial \boldsymbol{\xi}_{\boldsymbol{\theta}}}{\partial \boldsymbol{\theta}}$. The algorithm is in Appendix D.

**Experiment: control and planning.** Based on identified dynamics, we learn policies of each system to optimize a control objective with given $\boldsymbol{\theta}_{\text{obj}}$. We set loss (8) as the control objective (below called control loss). To parameterize policy (7), we use a Lagrange polynomial of degree $N$ (for planning) or neural network (for feedback control). iLQR [38] and guided policy search (GPS) [59] are compared. We set learning rate $\eta = 10^{-4}$ or $10^{-6}$ and run five trials for each system. Fig. 5a-5b are learning neural network feedback policies for the cart-pole and robot arm, respectively. The results show that PDP outperforms GPS for having lower control loss. Fig. 5c is motion planning for quadrotor using a polynomial policy. It shows that PDP achieves a competitive performance with iLQR. Compared to iLQR, PDP minimizes over polynomial policies instead of input sequences, and thus has a higher final loss which depends on the expressiveness of the polynomial: e.g., the polynomial of degree $N=35$ has a lower loss than that of $N=5$. Since iLQR can be viewed as '1.5-order' method (discussed in Section 2), it has faster converging speed than PDP which is only first-order, as shown in Fig. 5c. But iLQR is computationally extensive, PDP, instead, has a huge advantage of running time, as illustrated in Fig. 5d. Due to space constraint, we put detailed analysis between GPS and PDP in Appendix E.4.

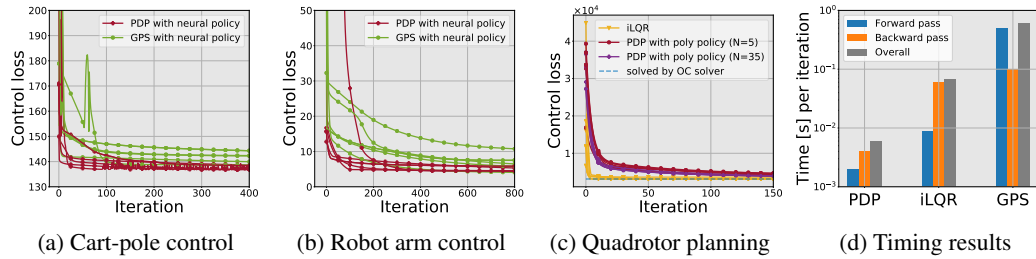

(a) Cart-pole control      (b) Robot arm control      (c) Quadrotor planning      (d) Timing results

Figure 5: (a-c) control loss v.s. iteration, (d) comparison for running time per iteration.

# 7   Discussion

**The related end-to-end learning frameworks.** Two lines of recent work are related to PDP. One is the recent work [60–64] that seeks to replace a layer within a deep neural network by an *argmin layer*, in order to capture the information flow characterized by a solution of an optimization. Similar to PDP, these methods differentiate the argmin layer through KKT conditions. They mainly focus on static optimization problems, which can not directly be applied to dynamical systems. The second line is the recent RL development [65–68] that embeds an implicit planner within a policy. The idea is analogous to MPC, because using a predictive OC system (i.e., embedded planner) to generate controls leads to better adaption to unseen situations. The key problem in these methods is to learn a planner (i.e., OC system), which is similar to our formulation. [65, 66] learn a path-integral OC system [69], which is a special class of OC systems. [68] learns an OC system in a latent space. However, all these methods adopt the 'unrolling' strategy to facilitate differentiation. Specifically, they treat the forward pass of solving an OC problem as an 'unrolled' computational graph of multiple steps of applying gradient descent, because by this computational graph, automatic differentiation tool [70] can be immediately applied. The drawbacks of this 'unrolling' strategy are apparent: (i) they need to store all intermediate results over the entire computational graph, thus are memory-expensive; and (ii) the accuracy of gradient depends on the length of the 'unrolled' graph, thus facing trade-off between complexity and accuracy. To address these, [67] develops a differentiable MPC framework, where in forward pass, a LQR approximation of the OC system is obtained, and in backward pass, the gradient is solved by differentiating such LQR approximation. Although promising, this framework has one main weakness: differentiating LQR requires to solve a large linear equation, which involves the inverse of a matrix of size $(2n+m)T \times (2n+m)T$, thus can incur huge cost when handling systems of longer horizons $T$. Detailed descriptions for all these methods is in Appendix F.

Compared to [35, 65–68], the efficiency of PDP stems from the following novel aspects. First, in forward pass, without needing an unrolled computational graph, PDP only computes and stores the resulting trajectory of the OC system, $\boldsymbol{\xi_\theta}$, (does not care about how $\boldsymbol{\xi_\theta}$ is solved). Second, without obtaining intermediate (LQR) approximations, PDP differentiates through PMP of the OC system to directly obtain the exact analytical gradient. Third, in the backward pass, unlike differentiable MPC which costs at least a complexity of $\mathcal{O}\left((m+2n)^2T^2\right)$ to differentiate a LQR approximation, PDP explicitly

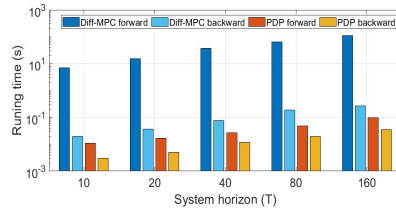

Figure 6: Runtime (per iteration) comparison between PDP and differentiable MPC for varying horizons of a pendulum system.

solves $\frac{\partial \boldsymbol{\xi_\theta}}{\partial \boldsymbol{\theta}}$ by an auxiliary control system, where thanks to the recursion structure, the memory and comptuation complexity of PDP is only $\mathcal{O}\left((m+2n)T\right)$. In Fig. 6, we have compared the running time of PDP with that of differentiable MPC. The results show PDP is 1000x faster than differentiable MPC. Due to space constraint, we put the detailed complexity analysis of PDP in Appendix G.

**Convergence and limitation of PDP.** Since all gradient quantities in PDP are analytical and exact, and the development of PDP does not involves any second-order derivative of functions or models, PDP essentially is a *first-order gradient-descent framework to solve non-convex bi-level optimization*. Therefore, in general, *PDP can only achieve local minima*. As explored by [71], if we pose further assumptions such as convexity and smoothness on all functions (dynamics, policy, loss, and control objective function), the global convergence of the bi-level programming could be established. But we do think these conditions are too restrictive for dynamical control systems. As a direction of future work, we will investigate the mild conditions for good convergence by taking advantage of control theory, e.g., Lyapunov theory. Due to space constraint, limitation of PDP is detailed in Appendix H.

# 8   Conclusions

This paper proposes a Pontryagin differentiable programming (PDP) methodology to establish an end-to-end learning framework for solving a range of learning and control tasks. The key contribution in PDP is that we incorporate the knowledge of optimal control theory as an inductive bias into the learning framework. Such combination enables PDP to achieve higher efficiency and capability than existing learning and control methods in solving many tasks including inverse reinforcement learning, system identification, and control/planning. We envision the proposed PDP could benefit to both learning and control fields for solving many high-dimensional continuous-space problems.

## Broader Impact

This work is expected to have the impacts on both learning and control fields.

- To the learning field, this work connects some fundamental topics in machine learning to their counterparts in the control field, and unifies some concepts from reinforcement learning, backpropagation/deep learning, and control theory in one generic learning framework. The contribution of this framework is a deep integration of optimal control theory into end-to-end learning process, leading to an optimal-control-informed end-to-end learning framework that is flexible enough to solve a broad range of learning and control tasks and efficient enough to handle high-dimensional and continuous-space problems. In a broad perspective, we hope that this paper could motivate more future work that integrates the benefits of both control and learning to promote efficiency and explainability of artificial intelligence.

- To the control field, this work proposes a generic paradigm, which shows how a challenging control task can be converted into a learning formulation and solved using readily-available learning techniques, such as (deep) neural networks and backpropagation. For example, the proposed framework, equipped with (deep) neural networks, shows significant advantage for handling non-linear system identification and optimal control over state-of-the-art control methods. Since classic control theory typically requires knowledge of models, we expect that this work could pave a new way to extend classic control with data-driven techniques.

Since the formulation of this paper does not consider the boundness or constraints of a decision-making system, the real-world use of this work on physical systems might possibly raise safety issues during the training process; e.g., the state or input of the physical system at some time instance might exceeds the safety bounds that are physically required. One option to address this is to include these safety boundness as soft constraints added to the control objective or loss that is optimized. In future work, we will formally discuss PDP within a safety framework.

## Acknowledgments and Disclosure of Funding

We acknowledge support for this research from Northrop Grumman Mission Systems' University Research Program.

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
