[Supplementary Material]

Supplementary materials for the Pontryagin Differentiable Programming paper

# A    Proof of Lemma 5.1

To prove Lemma 5.1, we just need to show that the Pontryagin's Maximum Principle for the auxiliary control system $\overline{\Sigma}(\boldsymbol{\xi_\theta})$ in (16) is exactly the differential PMP in (13). To this end, we define the following Hamiltonian for the auxiliary control system $\overline{\Sigma}(\boldsymbol{\xi_\theta})$:

$$\bar{H}_t = \mathrm{Tr}\left(\frac{1}{2}\begin{bmatrix}X_t\\U_t\end{bmatrix}'\begin{bmatrix}H_t^{xx} & H_t^{xu}\\H_t^{ux} & H_t^{uu}\end{bmatrix}\begin{bmatrix}X_t\\U_t\end{bmatrix} + \begin{bmatrix}H_t^{xe}\\H_t^{ue}\end{bmatrix}'\begin{bmatrix}X_t\\U_t\end{bmatrix}\right) + \mathrm{Tr}\left(\Lambda_{t+1}'(F_tX_t + G_tU_t + E_t)\right),\quad\text{(S.1)}$$

with $t = 0, 1, \cdots, T-1$. Here $\Lambda_{t+1} \in \mathbb{R}^{n \times r}$ denotes the costate (matrix) variables for the auxiliary control system. Based on Section 3 in [72], there exists a sequence of costates $\Lambda_{1:T}^{\boldsymbol{\theta}}$, which together the stationary solution $\{X_{0:T}^{\boldsymbol{\theta}}, U_{0:T-1}^{\boldsymbol{\theta}}\}$ to the auxiliary control system must satisfy the following the matrix version of PMP (we here follow the notation style used in (11)).

The dynamics equation:

$$\frac{\partial \bar{H}_t}{\partial \Lambda_{t+1}^{\boldsymbol{\theta}}} = \frac{\partial \mathrm{Tr}\left(\Lambda_{t+1}'(F_tX_t + G_tU_t + E_t)\right)}{\partial \Lambda_{t+1}}\Bigg|_{\substack{\Lambda_{t+1}=\Lambda_{t+1}^{\boldsymbol{\theta}}\\X_t=X_t^{\boldsymbol{\theta}}\\U_t=U_t^{\boldsymbol{\theta}}}}$$
$$= F_tX_t^{\boldsymbol{\theta}} + G_tU_t^{\boldsymbol{\theta}} + E_t = \mathbf{0}.\quad\text{(S.2a)}$$

The costate equation:

$$\frac{\partial \bar{H}_t}{\partial X_t^{\boldsymbol{\theta}}} = \frac{\partial \mathrm{Tr}\left(\frac{1}{2}X_t'H_t^{xx}X_t\right) + \partial \mathrm{Tr}\left(U_t'H_t^{ux}X_t\right) + \partial \mathrm{Tr}\left(H_t^{ex}X_t\right) + \partial \mathrm{Tr}\left(\Lambda_{t+1}'F_tX_t\right)}{\partial X_t}\Bigg|_{\substack{\Lambda_{t+1}=\Lambda_{t+1}^{\boldsymbol{\theta}}\\X_t=X_t^{\boldsymbol{\theta}}\\U_t=U_t^{\boldsymbol{\theta}}}}$$
$$= H_t^{xx}X_t^{\boldsymbol{\theta}} + H_t^{xu}U_t^{\boldsymbol{\theta}} + H_t^{xe} + F_t'\Lambda_{t+1}^{\boldsymbol{\theta}} = \Lambda_t^{\boldsymbol{\theta}}.\quad\text{(S.2b)}$$

Input equation:

$$\frac{\partial \bar{H}_t}{\partial U_t^{\boldsymbol{\theta}}} = \frac{\partial \mathrm{Tr}\left(\frac{1}{2}U_t'H_t^{uu}U_t\right) + \partial \mathrm{Tr}\left(U_t'H_t^{ux}X_t\right) + \partial \mathrm{Tr}\left(H_t^{eu}U_t\right) + \partial \mathrm{Tr}\left(\Lambda_{t+1}'G_tU_t\right)}{\partial U_t}\Bigg|_{\substack{\Lambda_{t+1}=\Lambda_{t+1}^{\boldsymbol{\theta}}\\X_t=X_t^{\boldsymbol{\theta}}\\U_t=U_t^{\boldsymbol{\theta}}}}$$
$$= H_t^{uu}U_t^{\boldsymbol{\theta}} + H_t^{ux}X_t^{\boldsymbol{\theta}} + H_t^{ue} + G_t'\Lambda_{t+1}^{\boldsymbol{\theta}} = \mathbf{0}.\quad\text{(S.2c)}$$

And boundary conditions:

$$\Lambda_T^{\boldsymbol{\theta}} = \frac{\partial \mathrm{Tr}(\frac{1}{2}X_T'H_T^{xx}X_T) + \partial \mathrm{Tr}((H_T^{xe})'X_T)}{\partial X_T}\Bigg|_{X_T=X_T^{\boldsymbol{\theta}}} = H_T^{xx}X_T^{\boldsymbol{\theta}} + H_T^{xe},\quad\text{(S.2d)}$$

and $X_0^{\boldsymbol{\theta}} = \mathbf{0}$. Note that in the above derivations, we used the following matrix calculus [72]:

$$\frac{\partial \mathrm{Tr}(AB)}{\partial A} = B',\quad \frac{\partial f(A)}{\partial A'} = \left[\frac{\partial f(A)}{\partial A}\right]',\quad \frac{\partial \mathrm{Tr}(X'HX)}{\partial X} = HX + H'X,\quad\text{(S.3)}$$

and the following matrix trace properties:

$$\mathrm{Tr}(A) = \mathrm{Tr}(A'),\quad \mathrm{Tr}(ABC) = \mathrm{Tr}(BCA) = \mathrm{Tr}(CAB),\quad \mathrm{Tr}(A+B) = \mathrm{Tr}(A) + \mathrm{Tr}(B).\quad\text{(S.4)}$$

Since the above obtained PMP equations (S.2) are the same with the differential PMP in (13), we thus can conclude that the Pontryagin's Maximum Principle of the auxiliary control system $\overline{\Sigma}(\boldsymbol{\xi_\theta})$ in (16) is exactly the differential PMP equations (13), and thus (17) holds. This completes the proof.    □

# B    Proof of Lemma 5.2

Based on Lemma 5.1 and its proof, we known that the PMP of the auxiliary control system, (S.2), is exactly the differential PMP equations (13). Thus below, we only look at the differential PMP equations in (S.2). From (S.2c), we solve for $U_t^{\boldsymbol{\theta}}$ (if $H_t^{uu}$ invertible):

$$U_t^{\boldsymbol{\theta}} = -(H_t^{uu})^{-1}\left(H_t^{ux}X_t^{\boldsymbol{\theta}} + G_t'\Lambda_{t+1}^{\boldsymbol{\theta}} + H_t^{ue}\right).\quad\text{(S.5)}$$

By substituting (S.5) into (S.2a) and (S.2b), respectively, and considering the definitions of matrices $A_t, R_t, M_t, Q_t$ and $N_t$ in (18), we have

$$X_{t+1}^{\theta} = A_t X_t^{\theta} - R_t \Lambda_{t+1}^{\theta} + M_t, \tag{S.6}$$

$$\Lambda_t^{\theta} = Q_t X_t^{\theta} + A_t' \Lambda_{t+1}^{\theta} + N_t, \tag{S.7}$$

for $t = 0, 1, \ldots, T-1$, and also the boundary condition in (S.2d)

$$\Lambda_T^{\theta} = H_T^{xx} X_T^{\theta} + H_T^{xe},$$

for $t = T$. Next, we prove that there exist matrices $P_t$ and $W_t$ such that

$$\Lambda_t^{\theta} = P_t X_t^{\theta} + W_t. \tag{S.8}$$

Proof by induction: (S.2d) shows that (S.8) holds for $t = T$ if $P_T = H_T^{xx}$ and $W_T = H_T^{xe}$. Assume (S.8) holds for $t+1$, then by manipulating (S.6) and (S.7), we have

$$\Lambda_t^{\theta} = \underbrace{\left(Q_t + A_t'(I + P_{t+1}R_t)^{-1}P_{t+1}A_t\right)}_{P_t} X_t^{\theta} + \underbrace{A_t'(I + P_{t+1}R_t)^{-1}(W_{t+1}+P_{t+1}M_t) + N_t}_{W_t}, \tag{S.9}$$

which indicates (S.8) holds for $t$, if $P_t$ and $W_t$ satisfy (18a) and (18b), respectively. Substituting (S.8) to (S.7) and also considering (S.5) will lead to (19a). (19b) directly results from (S.2a). We complete the proof. $\qquad \square$

## C   Proof of the Discrete-Time Pontryagin's Maximum Principle

We here provide an easy-approach derivation of the discrete-time PMP based on Karush-Kuhn-Tucker (KKT) conditions in non-linear optimization [73]. The original derivation for continuous optimal control systems uses the calculus of variation theory, which can be found in [39] and [74].

We view the optimal control system (1) with a fixed $\theta$ as a constrained optimization problem, where the objective function is given by $J(\theta)$ and the constraints given by dynamics $f(\theta)$. Define the following Lagrangian for this constrained optimization problem:

$$
\begin{aligned}
L &= J(\theta) + \sum_{t=0}^{T-1} \lambda_{t+1}' \big( f(x_t, u_t, \theta) - x_{t+1} \big) \\
&= \sum_{t=0}^{T-1} \left( c_t(x_t, u_t, \theta) + \lambda_{t+1}' \big( f(x_t, u_t, \theta) - x_{t+1} \big) \right) + h(x_T, \theta) \\
&= \sum_{t=0}^{T-1} \left( H_t - \lambda_{t+1}' x_{t+1} \right) + h(x_T, \theta),
\end{aligned}
\tag{S.10}
$$

where $\lambda_t$ is the Lagrange multiplier for the dynamics constraint for $t = 1, 2, \cdots, T$, and the third line in (S.10) is due to the definition of Hamiltonian in (10). According to the KKT conditions, for the optimal solution $\xi_{\theta} = \{x_{0:T}^{\theta}, u_{0:T-1}^{\theta}\}$, there must exist the multiplers $\lambda_{1:T}^{\theta}$ (in optimal control they are called costates) such that the following first-order conditions are satisfied:

$$\frac{\partial L}{\partial \lambda_{1:T}^{\theta}} = \mathbf{0}, \quad \frac{\partial L}{\partial x_{0:T}^{\theta}} = \mathbf{0}, \quad \frac{\partial L}{\partial u_{0:T-1}^{\theta}} = \mathbf{0}. \tag{S.11}$$

By extending the above three conditions in (S.11) at each $\lambda_t$, $x_t$ and $u_t$, respectively, and particularly taking care of $x_T$, we will obtain

$$\mathbf{0} = f(x_t^{\theta}, u_t^{\theta}; \theta) - x_{t+1}^{\theta}, \tag{S.12a}$$

$$\mathbf{0} = \frac{\partial H_t}{\partial x_t^{\theta}} - \lambda_t^{\theta} = \frac{\partial c_t}{\partial x_t^{\theta}} + \frac{\partial f'}{\partial x_t^{\theta}} \lambda_{t+1}^{\theta} - \lambda_t^{\theta}, \tag{S.12b}$$

$$\mathbf{0} = \frac{\partial H_t}{\partial u_t^{\theta}} = \frac{\partial c_t}{\partial u_t^{\theta}} + \frac{\partial f'}{\partial u_t^{\theta}} \lambda_{t+1}^{\theta}, \tag{S.12c}$$

$$\mathbf{0} = \frac{\partial h}{\partial x_T^{\theta}} - \lambda_T^{\theta}, \tag{S.12d}$$

respectively, which are exactly the PMP equations in (11). This completes the proof. $\qquad \square$

# D  Algorithms Details for Different Learning Modes

---

**Algorithm 2:** Solving $\frac{\partial \boldsymbol{\xi}_{\boldsymbol{\theta}}}{\partial \boldsymbol{\theta}}$ using Auxiliary Control System

---

**Input:** The trajectory $\boldsymbol{\xi}_{\boldsymbol{\theta}}$ generated by the system $\boldsymbol{\Sigma}(\boldsymbol{\theta})$

      Compute the coefficient matrices (14) to obtain the auxiliary control system $\overline{\boldsymbol{\Sigma}}(\boldsymbol{\xi}_{\boldsymbol{\theta}})$ in (16);

          **def** Auxiliary_Control_System_Solver ( $\overline{\boldsymbol{\Sigma}}(\boldsymbol{\xi}_{\boldsymbol{\theta}})$ ):    ▷ implementation of Lemma 5.2

              Set $P_T = H_T^{xx}$ and $W_T = H_T^{xe}$;

              **for** $t \leftarrow T$ **to** $0$ **by** $-1$ **do**

                Update $P_t$ and $W_t$ using equations (18);        ▷ backward in time

              **end**

              Set $X_0^{\boldsymbol{\theta}} = \mathbf{0}$;

              **for** $t \leftarrow 0$ **to** $T$ **by** $1$ **do**

                Update $X_t^{\boldsymbol{\theta}}$ and $U_t^{\boldsymbol{\theta}}$ using equations (19);        ▷ forward in time

              **end**

              **Return:** $\{X_{0:T}^{\boldsymbol{\theta}}, U_{0:T-1}^{\boldsymbol{\theta}}\}$

**Return:** $\frac{\partial \boldsymbol{\xi}_{\boldsymbol{\theta}}}{\partial \boldsymbol{\theta}} = \{X_{0:T}^{\boldsymbol{\theta}}, U_{0:T-1}^{\boldsymbol{\theta}}\}$

---

---

**Algorithm 3:** PDP Algorithm for IRL/IOC Mode

---

**Data :** Expert demonstrations $\{\boldsymbol{\xi}^{\mathrm{d}}\}$

**Parameterization:** The parameterized optimal control system $\boldsymbol{\Sigma}(\boldsymbol{\theta})$ in (1)

**Loss:** $L(\boldsymbol{\xi}_{\boldsymbol{\theta}}, \boldsymbol{\theta})$ in (4)

**Initialization :** $\boldsymbol{\theta}_0$, learning rate $\{\eta_k\}_{k=0,1,\cdots}$

**for** $k = 0, 1, 2, \cdots$ **do**

    Solve $\boldsymbol{\xi}_{\boldsymbol{\theta}_k}$ from the current optiaml control system $\boldsymbol{\Sigma}(\boldsymbol{\theta}_k)$ ;      ▷ using any OC solver

    Obtain $\frac{\partial \boldsymbol{\xi}_{\boldsymbol{\theta}}}{\partial \boldsymbol{\theta}}\big|_{\boldsymbol{\theta}_k}$ using Algorithm 2 given $\boldsymbol{\xi}_{\boldsymbol{\theta}_k}$ ;      ▷ using Algorithm 2

    Obtain $\frac{\partial L}{\partial \boldsymbol{\xi}}\big|_{\boldsymbol{\xi}_{\boldsymbol{\theta}_k}}$ from the given loss function $L(\boldsymbol{\xi}_{\boldsymbol{\theta}}, \boldsymbol{\theta})$ ;

    Apply the chain rule (9) to obtain $\frac{dL}{d\boldsymbol{\theta}}\big|_{\boldsymbol{\theta}_k}$ ;

    Update $\boldsymbol{\theta}_{k+1} \leftarrow \boldsymbol{\theta}_k - \eta_k \frac{dL}{d\boldsymbol{\theta}}\big|_{\boldsymbol{\theta}_k}$;

**end**

---

---

**Algorithm 4:** PDP Algorithm for SysID Mode

---

**Data:** Input-state data $\{\boldsymbol{\xi}^{\mathrm{o}}\}$

**Parameterization:** The parameterized dynamics model $\boldsymbol{\Sigma}(\boldsymbol{\theta})$ in (5)

**Loss:** $L(\boldsymbol{\xi}_{\boldsymbol{\theta}}, \boldsymbol{\theta})$ in (6)

**Initialization:** $\boldsymbol{\theta}_0$, learning rate $\{\eta_k\}_{k=0,1,\cdots}$

**for** $k = 0, 1, 2, \cdots$ **do**

    Obtain $\boldsymbol{\xi}_{\boldsymbol{\theta}_k}$ by iteratively integrating $\boldsymbol{\Sigma}(\boldsymbol{\theta}_k)$ in (5) for $t = 0, ..., T-1$;

    Compute the coefficient matrices (14) to obtain the auxiliary control system $\overline{\boldsymbol{\Sigma}}(\boldsymbol{\xi}_{\boldsymbol{\theta}})$ in (20);

    Obtain $\frac{\partial \boldsymbol{\xi}_{\boldsymbol{\theta}}}{\partial \boldsymbol{\theta}}\big|_{\boldsymbol{\theta}_k}$ by iteratively integrating $\overline{\boldsymbol{\Sigma}}(\boldsymbol{\xi}_{\boldsymbol{\theta}_k})$ in (20) for $t = 0, ..., T-1$;

    Obtain $\frac{\partial L}{\partial \boldsymbol{\xi}}\big|_{\boldsymbol{\xi}_{\boldsymbol{\theta}_k}}$ from the given loss function in (6);

    Apply the chain rule (9) to obtain $\frac{dL}{d\boldsymbol{\theta}}\big|_{\boldsymbol{\theta}_k}$ ;

    Update $\boldsymbol{\theta}_{k+1} \leftarrow \boldsymbol{\theta}_k - \eta_k \frac{dL}{d\boldsymbol{\theta}}\big|_{\boldsymbol{\theta}_k}$;

**end**

---

| **Algorithm 5:** PDP Algorithm for Control/Planning Mode |
|---|

**Parameterization:** The parameterized-policy system $\boldsymbol{\Sigma}(\boldsymbol{\theta})$ in (7)
**Loss:** $L(\boldsymbol{\xi_\theta}, \boldsymbol{\theta})$ in (8)
**Initialization:** $\boldsymbol{\theta}_0$, learning rate $\{\eta_k\}_{k=0,1,\cdots}$
**for** $k = 0, 1, 2, \cdots$ **do**

> Obtain $\boldsymbol{\xi}_{\boldsymbol{\theta}_k}$ by iteratively integrating $\boldsymbol{\Sigma}(\boldsymbol{\theta}_k)$ in (7) for $t = 0, ..., T-1$;
>
> Compute the coefficient matrices (14) to obtain the auxiliary control system $\overline{\boldsymbol{\Sigma}}(\boldsymbol{\xi_\theta})$ in (21);
>
> Obtain $\frac{\partial \boldsymbol{\xi_\theta}}{\partial \boldsymbol{\theta}}\big|_{\boldsymbol{\theta}_k}$ by iteratively integrating $\overline{\boldsymbol{\Sigma}}(\boldsymbol{\xi}_{\boldsymbol{\theta}_k})$ in (21) for $t = 0, ..., T-1$;
>
> Obtain $\frac{\partial L}{\partial \boldsymbol{\xi}}\big|_{\boldsymbol{\xi}_{\boldsymbol{\theta}_k}}$ from the given loss function $L(\boldsymbol{\xi_\theta}, \boldsymbol{\theta})$ in (8);
>
> Apply the chain rule (9) to obtain $\frac{dL}{d\boldsymbol{\theta}}\big|_{\boldsymbol{\theta}_k}$ ;
>
> Update $\boldsymbol{\theta}_{k+1} \leftarrow \boldsymbol{\theta}_k - \eta_k \frac{dL}{d\boldsymbol{\theta}}\big|_{\boldsymbol{\theta}_k}$;

**end**

**Additional comments: combining different learning modes.** In addition to using different learning modes to solve different types of problems, one can combine different modes in a single learning task. For example, when solving model-based reinforcement learning, one can call SysID Mode to first learn a dynamics model, then use the learned dynamics in Control/Planning Mode to obtain an optimal policy. In problems such as imitation learning, one can first learn a dynamics model using SysID Mode, then use the learned dynamics as the initial guess in IRL/IOC Mode. In forward pass of IOC/IRL Mode, one can call Control/Planning Mode to solve the OC system. For control and planning problems, the loss required in Control/Planning Mode can be learned using IOC/IRL Mode. In MPC-based learning and control [67], one can use the general formulation in (3) to learn a MPC controller, and then execute the MPC controller by calling Control/Planning Mode.

# E    Experiment Details

We have released the PDP source codes and different simulation environments/systems in this paper as two standalone packages, both of which are available at https://github.com/wanxinjin/Pontryagin-Differentiable-Programming. The video demos for some of the experiments are available at https://wanxinjin.github.io/posts/pdp.

## E.1    System/Environment Setup

**Quadrotor maneuvering control on *SE*(3).** We consider a quadrotor system maneuvering on *SE*(3) space (i.e. full position and full attitude space). The equation of motion of a quadrotor is given by:

$$
\begin{aligned}
\dot{\boldsymbol{p}}_I &= \dot{\boldsymbol{v}}_I, \\
m\dot{\boldsymbol{v}}_I &= m\boldsymbol{g}_I + \boldsymbol{f}_I, \\
\dot{\boldsymbol{q}}_{B/I} &= \frac{1}{2}\Omega(\boldsymbol{\omega}_B)\boldsymbol{q}_{B/I}, \\
J_B\dot{\boldsymbol{\omega}}_B &= \boldsymbol{M}_B - \boldsymbol{\omega} \times J_B\boldsymbol{\omega}_B.
\end{aligned}
\tag{S.13}
$$

Here, the subscriptions $_B$ and $_I$ denote that a quantity is expressed in the quadrotor's body frame and inertial (world) frame, respectively; $m$ is the mass of the quadrotor, respectively; $\boldsymbol{p} \in \mathbb{R}^3$ and $\boldsymbol{v} \in \mathbb{R}^3$ are the position and velocity vector of the quadrotor; $J_B \in \mathbb{R}^{3\times3}$ is the moment of inertia of the quadrotor with respect to its body frame; $\boldsymbol{\omega}_B \in \mathbb{R}^3$ is the angular velocity of the quadrotor; $\boldsymbol{q}_{B/I} \in \mathbb{R}^4$ is the unit quaternion [75] describing the attitude of quadrotor with respect to the inertial frame; $\Omega(\boldsymbol{\omega}_B)$ is

$$
\Omega(\boldsymbol{\omega}_B) = \begin{bmatrix} 0 & -\omega_x & -\omega_y & -\omega_z \\ \omega_x & 0 & \omega_z & -\omega_y \\ \omega_y & -\omega_z & 0 & \omega_x \\ \omega_z & \omega_y & -\omega_x & 0 \end{bmatrix}
\tag{S.14}
$$

and used for quaternion multiplication; $\boldsymbol{M}_B \in \mathbb{R}^3$ is the torque applied to the quadrotor; and $\boldsymbol{f}_I \in \mathbb{R}^3$ is the force vector applied to the quadrotor's center of mass (COM). The total force

magnitude $\|\boldsymbol{f}_I\| \in \mathbb{R}$ (along the z-axis of the body frame) and torque $\boldsymbol{M}_B = [M_x, M_y, M_z]$ are generated by thrusts $[T_1, T_2, T_3, T_4]$ of the four rotating propellers of the quadrotor, which can be written as

$$\begin{bmatrix} \|\boldsymbol{f}_I\| \\ M_x \\ M_y \\ M_z \end{bmatrix} = \begin{bmatrix} 1 & 1 & 1 & 1 \\ 0 & -l_w/2 & 0 & l_w/2 \\ -l_w/2 & 0 & l_w/2 & 0 \\ c & -c & c & -c \end{bmatrix} \begin{bmatrix} T_1 \\ T_2 \\ T_3 \\ T_4 \end{bmatrix}, \tag{S.15}$$

with $l_w$ being the wing length of the quadrotor and $c$ a fixed constant.

We define the state and input vectors of the quadrotor system as

$$\boldsymbol{x} = [\boldsymbol{p}' \quad \boldsymbol{v}' \quad \boldsymbol{q}' \quad \boldsymbol{\omega}']' \in \mathbb{R}^{13} \quad \text{and} \quad \boldsymbol{u} = [T_1 \quad T_2 \quad T_3 \quad T_4]' \in \mathbb{R}^4. \tag{S.16}$$

respectively. In design of the quadrotor's control objective function, to achieve *SE*(3) maneuvering control performance, we need to carefully design the attitude error. As used in [76], we define the attitude error between the quadrotor's current attitude $\boldsymbol{q}$ and the goal attitude $\boldsymbol{q}_g$ as

$$e(\boldsymbol{q}, \boldsymbol{q}_g) = \frac{1}{2}\operatorname{Tr}(I - R'(\boldsymbol{q}_g)R(\boldsymbol{q})), \tag{S.17}$$

where $R(\boldsymbol{q}) \in \mathbb{R}^{3\times 3}$ are the direction cosine matrix directly corresponding to $\boldsymbol{q}$ (see [75] for more details). Other error term in the control objective is the distance to the respective goal:

$$e(\boldsymbol{p}, \boldsymbol{p}_g) = \|\boldsymbol{p} - \boldsymbol{p}_g\|^2, \quad e(\boldsymbol{v}, \boldsymbol{v}_g) = \|\boldsymbol{v} - \boldsymbol{v}_g\|^2, \quad e(\boldsymbol{\omega}, \boldsymbol{\omega}_g) = \|\boldsymbol{\omega} - \boldsymbol{\omega}_g\|^2. \tag{S.18}$$

**Two-link robot arm.** The dynamics of a two-link robot arm can be found in [77, p. 171], where the state vector is $\boldsymbol{x} = [\boldsymbol{q}, \dot{\boldsymbol{q}}]'$ with $\boldsymbol{q} \in \mathbb{R}^2$ the vector of joint angles and $\dot{\boldsymbol{q}} \in \mathbb{R}^2$ the vector of joint angular velocities, and the control input $\boldsymbol{u} \in \mathbb{R}^2$ is the vector of torques applied to each joint.

**Dynamics discretization.** The continuous-time dynamics of all experimental systems in Table 2 are discretized using the Euler method: $\boldsymbol{x}_{t+1} = \boldsymbol{x}_t + \Delta \cdot \boldsymbol{f}(\boldsymbol{x}_t, \boldsymbol{u}_t)$ with the discretization interval $\Delta = 0.05\text{s}$ or $\Delta = 0.1\text{s}$.

**Simulation environment source codes.** We have made different simulation environments/systems in Table 2 as a standalone Python package, which is available at `https://github.com/wanxinjin/Pontryagin-Differentiable-Programming`. This environment package is easy to use and has user-friendly interfaces for customization.

## E.2 Experiment of Imitation Learning

**Data acquisition.** The dataset of expert demonstrations $\{\boldsymbol{\xi}^d\}$ is generated by solving an expert optimal control system with the expert's dynamics and control objective parameter $\boldsymbol{\theta}^* = \{\boldsymbol{\theta}_{\text{dyn}}^*, \boldsymbol{\theta}_{\text{dyn}}^*\}$ given. We generate a number of five trajectories, where different trajectories $\boldsymbol{\xi}^d = \{\boldsymbol{x}_{0:T}^d, \boldsymbol{u}_{0:T-1}^d\}$ have different initial conditions $\boldsymbol{x}_0$ and time horizons $T$ ($T$ ranges from 40 to 50).

**Inverse KKT method.** We choose the inverse KKT method [52] for comparison because it is suitable for learning objective functions for high-dimensional continuous-space systems. We adapt the inverse KKT method, and define the KKT loss as the norm-2 violation of the KKT condition (S.11) by the demonstration data $\boldsymbol{\xi}^d$, that is,

$$\min_{\boldsymbol{\theta}, \boldsymbol{\lambda}_{1:T}} \left( \left\| \frac{\partial L}{\partial \boldsymbol{x}_{0:T}}(\boldsymbol{x}_{0:T}^d, \boldsymbol{u}_{0:T-1}^d) \right\|^2 + \left\| \frac{\partial L}{\partial \boldsymbol{u}_{0:T-1}}(\boldsymbol{x}_{0:T}^d, \boldsymbol{u}_{0:T-1}^d) \right\|^2 \right), \tag{S.19}$$

where $\frac{\partial L}{\partial \boldsymbol{x}_{0:T}}(\cdot)$ and $\frac{\partial L}{\partial \boldsymbol{u}_{0:T-1}}(\cdot)$ are defined in (S.11) and $\boldsymbol{\theta} = \{\boldsymbol{\theta}_{\text{dyn}}, \boldsymbol{\theta}_{\text{dyn}}\}$. We minimize the above KKT-loss with respect to the unknown $\boldsymbol{\theta}$ and the costate variables $\boldsymbol{\lambda}_{1:T}$.

Note that to illustrate the inverse-KKT learning results in Fig. 3, we plot the imitation loss $L(\boldsymbol{\xi}_{\boldsymbol{\theta}}, \boldsymbol{\theta}) = \|\boldsymbol{\xi}^d - \boldsymbol{\xi}_{\boldsymbol{\theta}}\|^2$ instead of the KKT loss (S.19), because we want to guarantee that the comparison criterion is the same across different methods. Thus for each iteration $k$ in minimizing the KKT loss (S.19), we use the parameter $\boldsymbol{\theta}_k$ to compute the optimal trajectory $\boldsymbol{\xi}_{\boldsymbol{\theta}_k}$ and obtain the imitation loss.

**Neural policy cloning.** For the neural policy cloning (similar to [78]), we directly learn a neural-network policy $\boldsymbol{u} = \boldsymbol{\pi}_{\boldsymbol{\theta}}(\boldsymbol{x})$ from the dataset using supervised learning, that is

$$\min_{\boldsymbol{\theta}} \sum_{t=0}^{T-1} \|\boldsymbol{u}_t^d - \boldsymbol{\pi}_{\boldsymbol{\theta}}(\boldsymbol{x}_t^d)\|^2. \tag{S.20}$$

**Learning neural control objective function.** In Fig. 3d, we apply PDP to learn a neural objective function of the robot arm. The neural objective function is constructed as

$$J(\boldsymbol{\theta}) = V_{\boldsymbol{\theta}}(\boldsymbol{x}) + 0.0001\|\boldsymbol{u}\|^2, \tag{S.21}$$

with $V_{\boldsymbol{\theta}}(\boldsymbol{x})$ a fully-connected feed-forward network with `n-n-1` layers and `tanh` activation functions, i.e., an input layer with `n` neurons equal to the dimension of state, $n$, one hidden layer with `n` neurons and one output layer with 1 neuron. $\boldsymbol{\theta}$ is the neural network parameter. We separate the input cost from the neural network because otherwise it will cause instability when solving OC problems in the forward pass. Also, in learning the above neural objective function, we fix the dynamics because otherwise it will also lead to instability of solving OC.

In the comparing GAIL method [56], we use the following hyper-parameters: the policy network is a fully-connected feed-forward network with `n-400-300-m` layers and `relu` activation functions; the discriminator network is a `(n+m)-400-300-1` fully-connected feed-forward network with `tanh` and `sigmoid` activation functions; and the policy regularizer $\lambda$ is set to zero.

**Results and validation.** In Fig. S1, we show more detailed results of imitation loss versus iteration for three systems (cart-pole, robot arm, and quadrotor). On each system, we run five trials for all methods with random initial guess, and the learning rate for all methods is set as $\eta = 10^{-4}$. In Fig. S4, we validate the learned models (i.e., learned dynamics and learned control objective) by performing motion planning of each system in unseen settings. Specifically, we set each system with new initial state $\boldsymbol{x}_0$ and horizon $T$ and plan the control trajectory using the learned models, and we also show the corresponding true trajectory of the expert.

### E.3   Experiment of System Identification

**Data acquisition.** In the system identification experiment, we collect a total number of five trajectories from systems (in Table 2) with dynamics known, wherein different trajectories $\boldsymbol{\xi}^{\mathrm{o}} = \{\boldsymbol{x}^{\mathrm{o}}_{0:T}, \boldsymbol{u}_{0:T-1}\}$ have different initial conditions $\boldsymbol{x}_0$ and horizons $T$ ($T$ ranges from 10 to 20), with random inputs $\boldsymbol{u}_{0:T-1}$ drawn from uniform distribution.

**DMDc method.** The DMDc method [57], which can be viewed as a variant of Koopman theory [6], estimates a linear dynamics model $\boldsymbol{x}_{t+1} = A\boldsymbol{x}_t + B\boldsymbol{u}_t$, using the following least square regression

$$\min_{A,B} \sum\nolimits_{t=0}^{T-1} \|\boldsymbol{x}^{\mathrm{o}}_{t+1} - A\boldsymbol{x}^{\mathrm{o}}_t - B\boldsymbol{u}_t\|^2. \tag{S.22}$$

**Neural network baseline.** For the neural network baseline, we use a neural network $\boldsymbol{f}_{\boldsymbol{\theta}}(\boldsymbol{x}, \boldsymbol{u})$ to represent the system dynamics, where the input of the network is state and control vectors, and output is the state of next step. We train the neural network by minimizing the following residual

$$\min_{\boldsymbol{\theta}} \sum\nolimits_{t=0}^{T-1} \|\boldsymbol{x}^{\mathrm{o}}_{t+1} - \boldsymbol{f}_{\boldsymbol{\theta}}(\boldsymbol{x}^{\mathrm{o}}_t, \boldsymbol{u}_t)\|^2. \tag{S.23}$$

**Learning neural dynamics model.** In Fig. 4d, we compare the performance of PDP with Adam [58] for learning the same neural dynamics model for the robot arm system. Here, the neural dynamics model is a fully-connected feed-forward neural network with `(m+n)-(2m+2n)-n` layers and `tanh` activation functions, that is, an input layer with `(m+n)` neurons equal to the dimension of state, $n$, plus the dimension of control $m$, one hidden layer with `(2m+2n)` neurons and one output layer with `(n)` neurons. The learning rate for the PDP and the PyTorch Adam is both set as $\eta = 10^{-5}$.

**Results and validation.** In Fig. S2, we show more detailed results of SysID loss versus iteration for the three systems (cart-pole, robot arm, and quadrotor). On each system, we run five trials with random initial guess, and we set the learning rate as $\eta = 10^{-4}$ for all methods. In Fig. S5, we use the learned dynamics model to perform motion prediction of each system in unactuated conditions (i.e., $\boldsymbol{u}_t = \boldsymbol{0}$), in order to validate the effectiveness/correctness of the learned dynamics models.

### E.4   Experiment of Control/Planning

We use the dynamics identified in the system ID part, and the specified control objective function is set as weighted distance to the goal, as given in Table 2 ($\boldsymbol{\theta}_{\mathrm{obj}}$ is given). Throughout the optimal control/planning experiments, we use the time horizons $T$ ranging from 20 to 40.

**Learning neural network policies.** On the cart-pole and robot-arm systems (in Fig. 5a and Fig. 5b), we learn a feedback policy by minimizing given control objective functions. For both systems, we parameterize the policy using a neural network. Specifically, we use a fully-connected feed-forward neural network which has a layer structure of n-n-m with $\tanh$ activation functions, i.e., there is an input layer with n neurons equal to the dimension of state, one hidden layer with n neurons and one output layer with m neurons. The policy parameter $\boldsymbol{\theta}$ is the neural network parameter. We apply the PDP Control/Planning mode in Algorithm 5 and set the learning rate $\eta = 10^{-4}$. For comparison, we apply the guided policy search (GPS) method [59] (its deterministic version) to learn the same neural policy with the learning rate $\eta = 10^{-6}$ ($\eta$ in GPS is used to update the Lagrange multipliers for the policy constraint and we choose $\eta = 10^{-6}$ because it achieves the most stable results).

**Motion planning with Lagrange polynomial policies.** On the 6-DoF quadrotor, we use PDP to perform motion planning, that is, to find a control sequence to minimize the given control cost (loss) function. Here, we parameterize the policy $\boldsymbol{u}_t = \boldsymbol{u}(t, \boldsymbol{\theta})$ as $N$-degree Lagrange polynomial [80] with $N+1$ pivot points evenly populated over the time horizon, that is, $\{(t_0, \boldsymbol{u}_0), (t_1, \boldsymbol{u}_1), \cdots, (t_N, \boldsymbol{u}_N)\}$ with $t_i = iT/N$, $i = 0, \cdots, N$. The analytical form of the parameterized policy is

$$\boldsymbol{u}(t, \boldsymbol{\theta}) = \sum_{i=0}^{N} \boldsymbol{u}_i b_i(t) \qquad \text{with} \qquad b_i(t) = \prod_{0 \le j \le N, j \ne i} \frac{t - t_j}{t_i - t_j}. \qquad (S.24)$$

Here, $b_i(t)$ is called Lagrange basis, and the policy parameter $\boldsymbol{\theta}$ is defined as

$$\boldsymbol{\theta} = [\boldsymbol{u}_0, \cdots, \boldsymbol{u}_N]' \in \mathbb{R}^{m(N+1)}. \qquad (S.25)$$

The above Lagrange polynomial parameterization has been normally used in some trajectory optimization method such as [41, 81]. In this planning experiment, we have used different degrees of Lagrange polynomials, i.e., $N = 5$ and $N = 35$, respectively, to show how policy expressiveness can influence the final control loss (cost). The learning rate in PDP is set as $\eta = 10^{-4}$. For comparison, we also apply iLQR [38] to solve for the optimal control sequence.

**Results** In Fig. S3, we show the detailed results of control loss (i.e. the value of control objective function) versus iteration for three systems (cart-pole, robot arm, and quadrotor). For each system, we run five trials with random initial parameter $\boldsymbol{\theta}_0$. In Fig. S6, we apply the learned neural network policies (for cart-pole and robot arm systems) and the Lagrange polynomial policy (for quadrotor system) to simulate the corresponding system. For reference, we also plot the optimal trajectory solved by an OC solver [79] (which corresponds to the minimal control cost).

**Comments on the result comparison between GPS [59] and PDP.** In learning feedback policies, comparing the results obtained by the guided policy search (GPS) [59] and PDP in Fig. S3 and in Fig. S6, we have the following remarks.

(1) PDP outperforms GPS in terms of having lower control loss (cost). This can be seen in Fig. S3 and Fig. S6 (in Fig. S6, PDP results in a simulated trajectory which is closer to the optimal one than that of GPS). This can be understood from the fact that GPS considers the policy as constraint and updates it in a supervised learning step during the learning process. Although GPS aims to *simultaneously* minimize the control cost and the degree to which the policy is violated, it does not necessarily mean that before the learning researches convergence, when *strictly following* a pre-convergence control policy, the system will have a cost as minimal as it can possibly achieve.

(2) Instead, PDP adopts a different way to synchronize the fulfillment of policy constraints and the minimization of the control cost. In fact, throughout the entire learning process, PDP always guarantees that the policy constraint is perfectly respected (as the forward pass strictly follows the policy). Therefore, the core difference between PDP and GPS is that PDP does not simultaneously minimize two aspects—the policy violation and control cost, instead, it enforces that one aspect— policy—is always respected and only focuses on minimizing the other—control cost. The benefit of doing so is that at each learning step, the control cost for PDP is always as minimal as it can possibly achieve. This explains why PDP outperforms GPS in terms of having lower control cost (loss).

Figure S1: Experiments for PDP IRL/IOC Mode: imitation loss versus iteration. For each system, we run five trials starting with random initial guess $\boldsymbol{\theta}_0$, and the learning rate is $\eta = 10^{-4}$ for all methods. The results show a significant advantage of the PDP over the neural policy cloning and inverse-KKT [52] in terms of lower training loss and faster convergence speed. Please see Appendix Fig. S4 for validation. Please find the video demo at `https://youtu.be/awVNiCIJCfs`.

Figure S2: Experiments for PDP SysID Mode: SysID loss versus iteration. For each system, we run five trials with random initial guess $\boldsymbol{\theta}_0$, and set the learning rate $\eta = 10^{-4}$ for all methods. The results show a significant advantage of the PDP over neural-network dynamics and DMDc in terms of lower training loss and faster convergence speed. Please see Fig. S5 for validation. Please find the video demo at https://youtu.be/PAyBZjDD6OY.

Figure S3: Experiments for PDP Control/Planning Mode: control loss (i.e., objective function value) versus iteration. For the cart-pole (top panel) and robot arm (middle panel) systems, we learn neural feedback policies, and compare with the GPS method [59]. For the quadrotor system, we perform motion planning with a Lagrange polynomial policy (we use different degree $N$), and compare with iLQR and an OC solver [79]. The results show that for learning feedback control policies, PDP outperforms GPS in terms of having lower control loss (cost); and for motion planning, iLQR has faster convergence speed than PDP. Please find the video demo at https://youtu.be/KTw6TAigfPY.

(a) Cart-pole           (b) Robot arm           (c) Quadrotor

Figure S4: Validation for the imitation learning experiment in Fig. S1. We preform motion planing for each system in unseen conditions (new initial condition and new time horizon) using the learned models. Results show that compared to the neural policy cloning and inverse KKT [52], PDP result can accurately plan the expert's trajectory in unseen settings. This indicates PDP can accurately learn the dynamics and control objective, and has the better generality than the other two. Although policy imitation has lower imitation loss than inverse KKT, it has the poorer performance in planing. This is because with limited data, the cloned policy can be over-fitting, while the inverse KKT learns a cost function, a high-level representation of policies, thus has better generality to unseen conditions.

(a) Cart-pole           (b) Robot arm           (c) Quadrotor

Figure S5: Validation for the system identification experiment in Fig. S2. We perform motion prediction in unactuated conditions ($\boldsymbol{u} = 0$) using the learned dynamics. Results show that compared to neural-network dynamics training and DMDc, PDP can accurately predict the motion trajectory of each systems. This indicates the effectiveness of the PDP in identifying dynamics models.

(a) Cart-pole           (b) Robot arm           (c) Quadrotor

Figure S6: Simulation of the learned policies in the control and planning experiment in Fig. S3. Fig. S6a-S6b are the simulations of the learned neural feedback policies on the cart-pole and robot arm systems, respectively, where we also plot the optimal trajectory solved by an OC solver [79] for reference. From Fig. S6a-S6b, we observe that PDP results in a trajectory that is much closer to the optimal one than that of GPS; this implies that PDP has lower control loss (please check our analysis on this in Appendix E.4) than GPS. Fig. S6c is the planning results for the quadrotor system using PDP, iLQR, and an OC solver [79], where we have used different degrees of Lagrange polynomial policies in PDP. The results show that PDP can successfully plan a trajectory very close to the ground truth optimal trajectory. We also observe that the accuracy of the resulting trajectory depends on choice of the policy parameterization (i.e., expressive power): for example, the use of polynomial policy of a higher degree $N$ results in a trajectory closer to the optimal one (the one using the OC solver) than the use of a lower degree. iLQR is generally able to achieve high-accuracy solutions because it directly optimizes the loss function with respect to individual control inputs (instead of a parameterized policy), but this comes at the cost of high computation expense, as shown in Fig. 5d.

# F  Related End-to-End Learning Frameworks

As discussed in Section 7, two categories are related to this work. Here, we only detail the difference of PDP from the second category, i.e., the methods that learn an implicit planner within a RL policy.

**Differentiable MPC.** [67] develops an end-to-end differentiable MPC framework to jointly learn the system dynamics model and control objective function of an optimal control system. In the forward pass, it first uses iLQR [38] to solve the optimal control system and find a fixed point, and then approximate the optimal control system by a LQR at the fixed point. In the backward pass, the gradient is obtain by differentiating the LQR approximation. This process, however, may have two drawbacks: first, since the differentiation in the backward pass is conducted on the LQR approximation instead of on the original system, the obtained gradient thus may not be accurate due to discrepancy of approximation; and second, computing the gradient of the LQR approximation requires the inverse of a coefficient matrix, whose size is $(2n + m)T \times (2n + m)T$ with $n$ and $m$ state and action dimensions, respectively, $T$ the time horizon of the OC system, thus this will cause huge computational cost when handling the system of longer time horizon $T$.

Compared to differentiable MPC, the first advantage of the PDP framework is that the differentiation in the backward pass is directly performed on the parameterized optimal control system (by differentiating through PMP). Second, we develop the auxiliary control system in the backward pass of PDP, whose trajectory is exactly the gradient of the system trajectory in the forward pass. The gradient then is iteratively solved using the auxiliary control system by Lemma 5.2 (Algorithm 2). Those proposed techniques enables the PDP to have significant advantage in computational efficiency over differentiable MPC. To illustrate this, we have compare the algorithm complexity for both PDP and differentiable MPC in Table S1 and provide an experiment in Fig. S7.

Figure S7: Runtime (per iteration) comparison between the PDP and differentiable MPC [67] for different time horizons of a pendulum system. Note that y-axis is log-scale, and the runtime is averaged over 100 iterations. Both methods are implemented in Python and run on the same machine using CPUs. The results show that the PDP runs 1000x faster than differentiable MPC.

**Path Integral Network.** [65] and [66] develop a differentiable end-to-end framework to learn path-integral optimal control systems. Path-integral optimal control systems [69] however are a limited category of optimal control systems, where the dynamics is affine in control input and the control objective function is quadratic in control input. More differently, this path integral network is essentially an 'unrolling' method, which means that the forward pass of solving optimal control is extended as a graph of multiple steps of applying gradient descent, and the solution of the optimal control system is considered as the output of the final step of the gradient descent operations. Although the advantage of this unrolling (gradient descent) computational graph is that it can immediately take advantage of automatic differentiation techniques such as TensorFlow [70] to obtain the gradient in backpropagation, its drawback is however obvious: the framework is both memory- and computationally- expensive because it needs to store and traverse all intermediate results of the gradient descent process along the graph; furthermore, there is a conflict between computational complexity and accuracy in the forward pass. We have provided its complexity analysis in Table S1.

**Universal Planning Network**. In [68], the authors develop an end-to-end imitation learning framework consisting of two layers: the inner layer is a planner, which is formulated as an optimal control system in a latent space and is solved by gradient descent, and an outer layer to minimize the imitation

loss between the output of inner layer and expert demonstrations. However, this framework is also based on the 'unrolling' strategy. Specifically, the inner planning layer using gradient descent is considered as a large computation graph, which chains together the sub-graphs of each step of gradient descent. In the backward pass, the gradient derived from the outer layer back-propagates through the entire computation graph. Again, this unrolled learning strategy will incur huge memory and computation costs in implementation. Please find its complexity analysis in Table S1.

Different from the above 'unrolling' learning methods [65, 66, 68, 82], the proposed PDP method handles the learning of optimal control systems in a 'direct and compact' manner. Specifically, in forward pass, PDP only obtains and stores the final solution of the optimal control system and does not care about the (intermediate) process of how such solution is obtained. Thus, the forward pass of the PDP accepts any external optimal control solver such as CasADi [79]. Using the solution in the forward pass, the PDP then automatically builds the auxiliary control system, based on which, the exact *analytical* gradient is solved efficiently in backward pass. Such features guarantee that the complexity of the PDP framework is only linearly scaled up to the time horizon of the system, which is significantly efficient than the above 'unrolling' learning methods (please find the comparison in Table S1). In Appendix G, we will present the detailed complexity analysis.

Table S1: Complexity comparison for different end-to-end learning frameworks

| Learning frameworks | Forward pass | | Backward pass | |
|---|---|---|---|---|
| | Method and accuracy | Complexity (linear to) | Method | Complexity (linear to) |
| PI-Net [65] | $N$-step unrolled graph using gradient descent; accuracy depends on $N$ | computation: $NT$ memory: $NT$ | Back-propagation over the unrolled graph | computation: $NT$ memory: $NT$ |
| UPN [68] | $N$-step unrolled graph using gradient descent; accuracy depends on $N$ | computation: $NT$ memory: $NT$ | Back-propagation over the unrolled graph | computation: $NT$ memory: $NT$ |
| Diff-MPC [67] | iLQR finds fixed points; can achieve any accuracy | computation: — memory: $T$ | Differentiate the LQR approximation and solve linear equations | computation: $T^2$ memory: $T^2$ |
| PDP | Accept any OC solver; can achieve any accuracy | computation: —, memory: $T$ | Auxiliary control system | computation: $T$, memory: $T$ |

*Here $T$ denotes the time horizon of the system;

## G    Complexity of PDP

We consider the algorithm complexity of different learning modes of PDP (see Appendix D), and suppose that the time horizon of the parameterized system $\boldsymbol{\Sigma}(\boldsymbol{\theta})$ is $T$.

IRL/IOC Mode (Algorithm 3): in forward pass, PDP needs to obtain and store the optimal trajectory $\boldsymbol{\xi}_{\boldsymbol{\theta}}$ of the optimal control system $\boldsymbol{\Sigma}(\boldsymbol{\theta})$ in (1), and this optimal trajectory can be solved by any (external) optimal control solver. In backward pass, PDP first uses $\boldsymbol{\xi}_{\boldsymbol{\theta}}$ to build the auxiliary control system $\overline{\boldsymbol{\Sigma}}(\boldsymbol{\xi}_{\boldsymbol{\theta}})$ in (16) and then computes $\frac{\partial \boldsymbol{\xi}_{\boldsymbol{\theta}}}{\partial \boldsymbol{\theta}}$ by Lemma 5.2, which takes $2T$ steps.

SysID Mode (Algorithm 4): in forward pass, PDP needs to obtain and store the trajectory $\boldsymbol{\xi}_{\theta}$ of the original dynamics system $\boldsymbol{\Sigma}(\boldsymbol{\theta})$ in (5). Such trajectory is simply a result of iterative integration of (5), which takes $T$ steps. In backward pass, PDP first uses $\boldsymbol{\xi}_{\theta}$ to build the auxiliary control system $\overline{\boldsymbol{\Sigma}}(\boldsymbol{\xi}_{\theta})$ in (20) and then computes $\frac{\partial \boldsymbol{\xi}_{\theta}}{\partial \boldsymbol{\theta}}$ by iterative integration of (20), which takes $T$ steps.

Control/Planning Mode (Algorithm 5): in forward pass, PDP needs to obtain and store the trajectory $\boldsymbol{\xi}_{\theta}$ of the controlled system $\boldsymbol{\Sigma}(\boldsymbol{\theta})$ in (7). Such trajectory is simply a result of iterative integration of (7), which takes $T$ steps. In backward pass, PDP first uses $\boldsymbol{\xi}_{\theta}$ to build an auxiliary control system $\overline{\boldsymbol{\Sigma}}(\boldsymbol{\xi}_{\theta})$ in (21) and then computes $\frac{\partial \boldsymbol{\xi}_{\theta}}{\partial \boldsymbol{\theta}}$ by integration of (21), which takes $T$ steps.

Therefore, we can summarize that the memory- and computational- complexity for the PDP framework is only linear to the time horizon $T$ of the parameterized system $\boldsymbol{\Sigma}(\boldsymbol{\theta})$. This is significantly advantageous over existing end-to-end learning frameworks, as summarized in Table S1.

# H  Limitation of PDP

**PDP is a first-order algorithm.** We observe that (i) all gradient quantities in PDP are analytical and exact; (ii) the development of PDP does not involve any second-order derivative/approximation of functions or models (note that PMP is a first-order optimality condition for optimal control); and (iii) PDP minimizes a loss function directly with respect to unknown parameters in a system using gradient descent. Thus, we conclude that PDP is a first-order gradient-descent based optimization framework. Specifically for the SysID and Control/Planning modes of PDP, they are also first-order algorithms. When using these modes to solve optimal control problems, this first-order nature may bring disadvantages of PDP compared to high-order methods, such as iLQR which can be considered as 1.5-order because it uses second-order derivative of a value function and first-order derivative of dynamics, or DDP which is a second-order method as it uses the second-order derivatives of both value function and dynamics. The disadvantages of PDP have already been empirically shown in Fig. 5c and Fig. S3, where the converging speed of PDP in its planning mode is slower than that of iLQR. For empirical comparisons between first- and second-order techniques, we refer the reader to [83].

**Convergence to local minima.** *Since PDP is a first-order gradient-descent based algorithm, PDP can only achieve local minima for general non-convex optimization problems in (3).* Furthermore, we observe that the general problem in (3) belongs to a bi-level optimization framework. As explored in [71], under certain assumptions such as convexity and smoothness on models (e.g., dynamics model, policy, loss function and control objective function), global convergence of the bi-level optimization can be established. But we think such conditions are too restrictive in the context of dynamical control systems. As a future direction, we will investigate mild conditions for good convergence by resorting to dynamical system and control theory, such as Lyapunov theory.

**Parameterization matters for global convergence.** Although PDP only achieves local convergence, these still exists a question of how likely PDP can obtain the global convergence. In our empirical experiments, we find that how models are parameterized matters for good convergence performance. For example, in IOC/IRL mode, we observe that using a neural network control objective function (in Fig. 3d) is more likely to get trapped in local minima than using the parameterization of weighted distance objective functions (in Fig. 3a-3c). In control/planning mode, using a deeper neural network policy (in Fig. 5a-5b) is more like to result in local minima than using a simpler one. Also in the motion planning experiment, we use the Lagrange polynomial to parameterize a policy instead of using standard polynomials, because the latter can lead to poor conditioning and sensitivity issues (a small change of polynomial parameter results in large change in performance) and thus more easily get stuck in local minima. One high-level explanation is that more complex parameterization will bring extreme non-convexity to the optimization problem, making the algorithm more easily trapped in local minima. Again, how to theoretically justify those empirical experience and find the mild conditions for global convergence guarantee still needs to be investigated in future research.

# I  PDP to Solve 6-DoF Rocket Powered Landing Problems

As a final part in this supplementary, we will demonstrate the capability of PDP to solve the more challenging 6-DoF rocket powered landing problems.

We here omit the description of mechanics modeling for the 6-DoF powered rocket system, and refer the reader to Page 5 in [84] for the rigid body dynamics model of a rocket system (the notations and coordinates used below follows the ones in [84]). The state vector of the rocket system is defined as

$$\boldsymbol{x} = \begin{bmatrix} m & \boldsymbol{r}'_{\mathcal{I}} & \boldsymbol{v}'_{\mathcal{I}} & \boldsymbol{q}'_{\mathcal{B}/\mathcal{I}} & \boldsymbol{\omega}'_{\mathcal{B}} \end{bmatrix}' \in \mathbb{R}^{14}, \tag{S.26}$$

where $m \in \mathbb{R}$ is the mass of the rocket; $\boldsymbol{r}_{\mathcal{I}} \in \mathbb{R}^3$ and $\boldsymbol{v}_{\mathcal{I}} \in \mathbb{R}^3$ are the position and velocity of the rocket (center of mass) in the inertially-fixed Up-East-North coordinate frame; $\boldsymbol{q}_{\mathcal{B}/\mathcal{I}} \in \mathbb{R}^4$ is the unit quaternion denoting the attitude of rocket body frame with respect to the inertial frame (also see the description in the quadrotor dynamics in Appendix E.1); and $\boldsymbol{\omega}_{\mathcal{B}} \in \mathbb{R}^3$ is the angular velocity of the rocket expressed in the rocket body frame. In our simulation, we only focus on the final descending phase before landing, and thus assume the mass depletion during such a short phase is very slow and thus $\dot{m} \approx 0$. We define the control input vector of the rocket, which is the thrust force vector

$$\boldsymbol{u} = \boldsymbol{T}_{\mathcal{B}} = [T_x, T_y, T_z]' \in \mathbb{R}^3, \tag{S.27}$$

acting on the gimbal point of the engine (situated at the tail of the rocket) and is expressed in the body frame. Note that the relationship between the total torque $\boldsymbol{M}_{\mathcal{B}}$ applied to the rocket and the thrust force vector $\boldsymbol{T}_{\mathcal{B}}$ is $\boldsymbol{M}_{\mathcal{B}} = \boldsymbol{r}_{\mathcal{I},\mathcal{B}} \times \boldsymbol{T}_{\mathcal{B}}$, with $\boldsymbol{r}_{\mathcal{I},\mathcal{B}} \in \mathbb{R}^3$ being constant position vector from the center-of-mass of the rocket to the gimbal point of the engine. The continuous dynamics is discretized using the Euler method: $\boldsymbol{x}_{t+1} = \boldsymbol{x}_t + \Delta \cdot \boldsymbol{f}(\boldsymbol{x}_t, \boldsymbol{u}_t)$ with the discretization interval $\Delta = 0.1$s.

For the rocket system, the unknown dynamics parameter, $\boldsymbol{\theta}_{\text{dyn}}$, includes the rocket's initial mass $m_0$, and the moment of inertia $\boldsymbol{J}_{\mathcal{B}} \in \mathbb{R}^{3 \times 3}$, and the rocket length $\ell$, thus, $\boldsymbol{\theta}_{\text{dyn}} = \{m_0, \boldsymbol{J}_{\mathcal{B}}, \ell\} \in \mathbb{R}^8$.

For the control objective (cost) function, we consider a weighted combination of the following aspects:

- distance of the rocket position from the target position, associated with weight $w_1$;
- distance of the rocket velocity from the target velocity, associated with weight $w_2$;
- penalty of the excessive title angle of the rocket, associated with weight $w_3$;
- penalty of the side effects of the thrust vector, associated with weight $w_4$;
- penalty of the total fuel cost, associated with weighted $w_5$.

So the parameter of the control objective function, $\boldsymbol{\theta}_{\text{obj}} = [w_1, \; w_2, \; w_3, \; w_4, \; w_5]' \in \mathbb{R}^5$. In sum, the overall parameter for the 6-DoF rocket powered landing control system is

$$\boldsymbol{\theta} = \{\boldsymbol{\theta}_{\text{dyn}}, \; \boldsymbol{\theta}_{\text{obj}}\} \in \mathbb{R}^{13}. \tag{S.28}$$

**Imitation learning.** We apply the IRL/IOC mode of PDP to perform imitation learning of the 6-DoF rocket powered landing. The experiment process is similar to the experiments in Appendix E.2, where we collect five trajectories from an expert system with dynamics and control objective function both known (different trajectories have different time horizons $T$ ranging from 40 to 50 and different initial state conditions). Here we minimize imitation loss $L(\boldsymbol{\xi_\theta}, \boldsymbol{\theta}) = \|\boldsymbol{\xi}^{\text{d}} - \boldsymbol{\xi_\theta}\|^2$ over the parameter of dynamics and control objective, $\boldsymbol{\theta}$ in (S.28). The learning rate is set to $\eta = 10^{-4}$, and we run five trials with random initial parameter guess $\boldsymbol{\theta}_0$. The imitation loss $L(\boldsymbol{\xi_\theta}, \boldsymbol{\theta})$ versus iteration is plotted in Fig. S8a. To validate the learned models (the learned dynamics and the learned objective function), we use the learned models to perform motion planing of rocket powered landing in unseen settings (here we use new initial condition and new time horizon). The planing results are plotted in Fig. S8b, where we also plot the ground truth for comparison.

**System identification.** We apply the SysID mode of PDP to identify the dynamics parameter $\boldsymbol{\theta}_{\text{dyn}}$ of the rocket. The experiment process is similar to the experiments in Appendix E.3, where we collect five trajectories with different initial state conditions, time horizons ($T$ ranges from 10 to 20), and random control inputs. We minimize the SysID loss $L(\boldsymbol{\xi_\theta}, \boldsymbol{\theta}) = \|\boldsymbol{\xi}^{\text{o}} - \boldsymbol{\xi_\theta}\|^2$ over $\boldsymbol{\theta}_{\text{dyn}}$ in (S.28). The learning rate is set to $\eta = 10^{-4}$, and we run five trials with random initial parameter guess for $\boldsymbol{\theta}_{\text{dyn}}$. The SysID loss $L(\boldsymbol{\xi_\theta}, \boldsymbol{\theta})$ versus iteration is plotted in Fig. S9a. To validate the learned dynamics, we use it to predict the motion of rocket given a new sequence of control inputs. The prediction results are in Fig. S9b, where we also plot the ground truth for reference.

**Optimal powered landing control.** We apply the Control/Planning mode of PDP to find an optimal control sequence for the rocket to perform a successful powered landing. The experiment process is similar to the experiments performed for the quadrotor system in Appendix E.4. We set the time horizon as $T = 50$, and randomly choose an initial state condition $\boldsymbol{x}_0$ for the rocket. We minimize the control loss function, which is now a given control objective function with $\boldsymbol{\theta}_{\text{obj}}$ known. The control policy we use here is parameterized as the Lagrangian polynomial, as described in (S.24) in Appendix E.4, here with degree $N = 25$. The control loss is set as the control objective function learned in the previous imitation learning experiment. The learning rate is set to $\eta = 10^{-4}$, and we run five trials with random initial guess of the policy parameter. The the control loss $L(\boldsymbol{\xi_\theta}, \boldsymbol{\theta})$ versus iteration is plotted in Fig. S10a. To validate the learned optimal control policy, we use it to simulate the motion (control trajectory) of the rocket landing, and compare with the ground truth optimal trajectory obtained by an OC solver. The validation results are in Fig. S10b.

(a) Training

(b) Validation
($T_x$ is defined along the rocket direction)

Figure S8: (a) Training process for imitation learning of 6-DoF rocket powered landing: the imitation loss versus iteration; here we have performed five trials (labeled by different colors) with random initial parameter guess. (b) Validation: we use the learned models (dynamics and control objective function) to perform motion planning of the rocket powered landing in unseen settings (i.e. given new initial state condition and new time horizon requirement); here we also plot the ground-truth motion planning of the expert for reference. The results in (a) and (b) show that the PDP can accurately learn the dynamics and control objective function from demonstrations, and have good generalizability to novel situations. Please find the video demo at https://youtu.be/4RxDLxUcMp4.

(a) Training

(b) Validation

Figure S9: (a) Training process for identification of rocket dynamics: SysID loss versus iteration; here we have performed five trials (labeled by different colors) with random initial parameter guess. (b) Validation: we use the learned dynamics model to perform motion prediction of the rocket given a new control sequence; here we also plot the ground-truth motion (where we know the exact dynamics). The results in (a) and (b) show that the PDP can accurately identify the dynamics model of the rocket.

(a) Training

(b) Validation
($T_x$ is defined along the rocket)

Figure S10: (a) Training process of learning the optimal control policy for rocket powered landing: the control loss versus iteration; here we have performed five trials (labeled by different colors) with random initial guess of the policy parameter. (b) Validation: we use the learned policy to simulate the rocket control trajectory; here we also plot the ground-truth optimal control solved by an OC solver. The results in (a) and (b) show that the PDP can successfully find the optimal control policy (or optimal control sequence) to successfully perform the rocket powered landing. Please find the video demo at https://youtu.be/5Jsu772Sqcg.