[Reviews · NeurIPS 2020]

Review 1

Summary and Contributions: The authors propose an end-to-end learning methodology to be applied to many different domains such as system identification, reinforcement learning (RL) / optimal control (OC) and inverse RL/OC methods. This is possible by means of an unconstrained nonlinear formulation of a generic problem parameterized in terms of $\theta$ both in the dynamics and in the cost. The learning problem is then formulated by means of a suitable loss function for a specific application, and optimized with respect to such $\theta$. In order to backpropagate through the loss, the gradient computation is solved by using Pontryagin's Maximum Principle.

Strengths: The key contribution of this work consists in the use of Pontryagin's Maximum Principle (PMP) to compute the trajectory gradient, needed to backpropagate through the end-to-end learning scheme. The gradient can be effectively computed by using the fact that the optimal trajectory should obey the optimality conditions derived from the Hamiltonian, meaning that the trajectory gradient can be expressed in terms of the gradients of these conditions. As a further step, the optimality condition gradients formulate a new differential PMP, which in turn can be seen as the conditions of a new artificial optimal control problem modeled as an unconstrained quadratic program. This basically means that the gradient computation of a trajectory determined by a nonlinear problem is solved by means of an auxiliary artificial linear problem that can be easily solved via standard recursive procedure for solving finite-time horizon LQR problems, obtaining an elegant result.

Weaknesses: While the end-to-end learning framework should be brought into focus as it is the core component of this paper, I have the impression that the authors reserve a lot more space to an (arguable) list of methods to which such framework can be applied. Most of the introduction and the related work section focus on these different methodologies rather than drawing comparisons to other end-to-end learning frameworks. Furthermore, the rather ambitious goal of covering so many different fields clearly cannot be accomplished in just a page of background material. The whole learning dynamics literature misses Bayesian models (e.g. linear in the parameters), feature extraction, Markov decision processes, graphical models, Gaussian processes etc. and boils it down to mere (deep) neural networks. Similarly, the learning optimal policies sections forgets all the model-based control literature such as model predictive control. The choice of broadening so much the spectrum has two main counter effects: - It shifts the focus of the paper, rather than highlighting the central contribution, which furthermore contains the novelty. - It oversimplifies so many different research areas, resulting in a poor literature review. I would strongly suggest to both narrow down and shift the focus of the paper.

Correctness: The method seems to be correct, although I am confused by the "Control/Planning mode". Can't this be directly solved by any model-based controller (MPC or LQR) if the dynamics are known? Also, I am confused by the imitation learning results: it seems that the neural policy is consistently exceeding the inverse KKT in the learning phase, but then in the planning phase it performs very poorly. Could this behavior be clarified?

Clarity: The paper is well written and clear.

Relation to Prior Work: As previously mentioned, I think the paper lacks some of the other end-to-end learning approaches. Particularly related to this paper is e.g. "Differentiable MPC for End-to-end Planning and Control" B. Amos et al.

Reproducibility: Yes

Additional Feedback: ----------------------------------------- EDIT -------------------------------------------------- Unfortunately, the author response did not convince me. My concern is not really about the method explained in Sections 4 and 5 which is, as I already state in my review, an interesting formulation. I am mostly concerned with the background and the scope: the authors are tackling a simple nonlinear unconstrained problem, which in the control/planning mode could potentially be solved by a PID controller. Given that the authors claim to have identified a model of the system, also MPC is an established method. The argument of needing a solver to find a solution is a bit weak since even interior point methods are based on gradient computations, which the authors themselves use to backpropagate their end-to-end framework. The extension to system identification is a bit stretched, given that there are a number of tailored research streams trying to solve similar problems: I feel that the comparison to a neural network as a representation of the whole sys-id community is a bit reductive. Probably the IOC interpretation is the best fit, even though the results show very similar behavior to inverse KKT, which in essence is very similar to what the authors are doing, and can additionally handle constraints, which are present in most real-world problems. I strongly believe that there are a number of tailored methods for each of the mentioned “modes” that can potentially do better than a generic end-to-end framework. The focus should rather be shifted to cases in which the dynamics/cost is hard to identify/control due to inherent application complexity, which does not allow established methods to be used. In such case, also the experimental section would gain much more strength. For these reasons I decide to keep my score.


Review 2

Summary and Contributions: This paper introduces Pontryagin differentiable programming, a framework for end-to-end learning and control. The Pontryagin approach (based on the calculus of variations) to optimal control optimises over trajectories, as opposed to the more commonly used HJB fomulation. The key contribution of this work is to show that by differentiating and reformulating the necessary conditions for optimal control, it is possible to derive a so-called auxilliary control system that can be solved using a standard optimal control solver. This allows for relatively simple trajectory gradient calculation and optimisation over entire trajectories. Results show that this is effective in a number of settings (imitation learning, control and system identification) and computationally more efficient than common approaches like iLQR.

Strengths: I particularly enjoyed reading this paper. The formulation is applicable to a wide range of problem settings, and I believe that the NeurIPS community will benefit from a reminder of Pontryagin's minimum principle, and the fact that HJB is not the only approach to solve model-based RL problems. I believe that the idea of differentiating the necessary conditions for optimal control is new to the control community, and the discovery of the auxiliary control system is particularly useful, as this can be solved using standard LQR recursions. Comparisons with diff-mpc (ilqr) show clear computational benefits of this approach. Experiments are detailed, covering a broad range of settings and the supplementary material is particularly well laid out and thoughtful.

Weaknesses: For many of the experiments considered here, trajectories are parameterised using high degree polynomials. I would have liked to have seen more experiments where the trajectories are parameterised using a suitable neural policy, and PMP is still used for learning. One of the main benefits of end-to-end learning systems is that it allows gradients to be propagated further, eg. through perception networks in order to allow control from high dimensional sensory information. It is unclear whether the proposed framework naturally allows for this, and if so, how this would be implemented? My experience with shooting-like methods has been that they struggle to converge for more complex optimisation settings and costs, and I believe that the proposed approach may be vulnerable to these problems, if for example, used to optimise a neural policy as opposed to the simpler polynomial trajectories. I would have liked to see more discussion around limitations in convergence - under what conditions do you expect the framework to fail? Since the system relies on an OC solver in the middle, what happens if the OC solver fails to find a solution? Are there any convergence guarantees here?

Correctness: I believe that the claims are correct, and the methodology is clear.

Clarity: The paper is clearly laid out, and the explanation of the framework is well structured.

Relation to Prior Work: Relation to prior work is clearly discussed, although some of the more interesting discussions (comparisons with differentiable MPC and path integral networks) are in the supplementary material and not in the primary text.

Reproducibility: Yes

Additional Feedback: I enjoyed reading this paper, and would be keen to try this out - will a code implementation be made available? Would you be able to comment on practical aspects of using the framework to train a neural policy conditioned on additional information, instead of a higher order polynomial? Can you comment on convergence and potential sources of failure in the optimisation process? *** Post rebuttal comments *** Thank you for responding to my reviews, as mentioned previously I believe this is a strong submission and look forward to exploring this work further.


Review 3

Summary and Contributions: EDIT after author response: The authors sufficiently addressed my major concerns about explicit computation of the Jacobian and the clarity of Sections 2-3, so I have changed my score from 6 --> 7. Per the points raised in R1's review, however, I think this paper could be further strengthened by focusing on cases that are not well-addressed by methods in the existing RL/OC literature, in addition to simply moving content from the appendix to the main paper. ------------------------ This paper proposes a unified framework for solving inverse RL/inverse optimal control, system identification, and control/planning problems, which employs implicit differentiation through Pontryagin’s Maximum Principle. In particular, the differential system is shown to be yet another control problem that can be solved in the backward pass. The paper applies this framework to imitation learning, system identification, and optimal control tasks in four experimental domains, and shows improved performance compared to other methods.

Strengths: The unified framing of IRL/IOC, SysID, and Control/Planning as problems differing only by parameter locations, control objectives, and loss functions is interesting, informative, and (to the best of my knowledge) novel, both conceptually and in terms of implementation. Differentiation through the PMP conditions and the framing of this differentiation as a “reverse mode” optimal control problem is also interesting, novel, and well-described. The experiments demonstrate the strength and generality of the proposed framework in tackling a variety of kinds of tasks often considered in reinforcement learning and optimal control. Overall, these contributions are well-executed and add significantly to the literature by unifying concepts from reinforcement learning, control theory, and backpropagation/deep learning.

Weaknesses: The presentation of the unified framework suffers from some flaws in the mathematical exposition, and could be greatly improved. In particular, the casual presentation of Equation (2) makes it difficult to parse. This casual presentation also (a) does not make it obvious that the later equations (e.g. Eq (3)) are actually bi-level optimization frameworks, which is what makes the contribution of the proposed paper difficult/novel, and (b) makes some of the definitions building off of Eq (2) also imprecise or hard to understand. Please see the “additional feedback” section for specific suggestions as to how to clarify this framework. As this conceptual framework is a major contribution of the paper, it is critical that the authors improve the presentation. While the PMP differentiation is novel, in most differentiation frameworks, it is rarely encouraged to explicitly compute the Jacobians (e.g. d\xi / d\theta) due to space inefficiency. Instead, it is desirable to compute dL/d\theta directly. For instance, Neural ODEs [1] propose backpropagation through a “forward” ODE via a “backward/adjoint” ODE (similar to the forward and backward OC systems in the present work); however, they directly compute the gradient of the loss with respect to the parameters, rather than calculating intermediate Jacobians (see the appendix of [1]). As another example, OptNet [2] proposes a clever way to directly compute dL/d\theta via a single linear system solve, derived via implicit differentiation through the KKT conditions (see Eq (7)--(8) of [2]). It is extremely important that the authors either compute the gradient of the loss with respect to the parameters, or explain why this is not needed/desirable/possible. While I do think the overall contributions of this paper are rather strong, I see the above limitations as being extremely important for the authors to address in order for this work to achieve its full impact. An additional (more minor) weakness is that timing results are not included. The authors compare different methods in the experimental results on the basis of iterations, but the iterations for the proposed PDP framework are likely more expensive. It is important that either these timing results be reported, or that the authors comment qualitatively on the expense of backpropagation and gradient search using the PDP layer. [1] Chen, Ricky TQ, et al. "Neural ordinary differential equations." Advances in neural information processing systems. 2018. [2] Amos, Brandon, and J. Zico Kolter. "Optnet: Differentiable optimization as a layer in neural networks." arXiv preprint arXiv:1703.00443 (2017).

Correctness: In general, the claims and method seem correct.

Clarity: I think Section 3 could be vastly improved (see additional feedback below). The descriptions in Section 2 could be condensed/made cleaner to make it clear that learning dynamics/learning optimal policies/learning control objective functions are all simply instantiations of the same problem but with different unknown parameters, different control objectives, and different losses (in order to hint at the framework in Section 3). The italicized points about each problem being viewed as gradient steps on some aspect of the system are not immediately clear without this context (which only comes up later in the paper). From Section 4 onwards, the paper is very clear and well-written.

Relation to Prior Work: The relevant previous work in RL and control seems well-described. However, the authors should discuss and cite the body of work in differentiable optimization layers for neural networks, particularly those papers that employ implicit differentiation through the KKT conditions of optimization problems (i.e, the optimization analogues of the PMP conditions in optimal control). Some examples include: [2] cited above [3] Donti, Priya, Brandon Amos, and J. Zico Kolter. "Task-based end-to-end model learning in stochastic optimization." Advances in Neural Information Processing Systems. 2017. [4] Wang, Po-Wei, et al. "SATNet: Bridging deep learning and logical reasoning using a differentiable satisfiability solver." arXiv preprint arXiv:1905.12149 (2019). [5] Wilder, Bryan, Bistra Dilkina, and Milind Tambe. "Melding the data-decisions pipeline: Decision-focused learning for combinatorial optimization." Proceedings of the AAAI Conference on Artificial Intelligence. Vol. 33. 2019. [6] de Avila Belbute-Peres, Filipe, et al. "End-to-end differentiable physics for learning and control." Advances in Neural Information Processing Systems. 2018.

Reproducibility: Yes

Additional Feedback: I would strongly suggest the following changes to Section 3: * Present the definition of \xi_\theta via an argmin, e.g. for Equation (2): \xi_\theta \in \argmin_{trajectories} J(\theta) s.t. x_0 = x^0, x_{t+1} = f(x_t, u_t, \theta) \forall t * Use the above definition in place of language such as “is generated by \Sigma(\theta)” or “\Sigma(\theta) will produce”, and make it clear that some algorithm — not the system — is producing these trajectories. * Remove the framing of \Sigma(\theta) as a “configurable box,” as this evokes language from implicit differentiation layers rather than implying that \Sigma(\theta) is a mathematical/theoretical construct that can be modified for IRL vs. SysID vs. planning. Instead, perhaps mention we change the precise details of \Sigma(\theta) for each of these settings. * Further clarify the difference between \xi^d and \xi^o given in the IRL vs. SysID definitions to mention that \xi^d is optimal with respect to the control objective and dynamics, where as \xi^o is not necessarily. (This exposition can be facilitated by using the argmin definition of \xi suggested above. In SysID mode, to clarify the point about externally-supplied outputs, constraints u^\theta_{0:T-1} = u_{0:T-1} could also be added.), Reproducibility: While the major theoretical results are reproducible, code for the experiments (or a detailed description of relevant algorithms and hyperparameters) would be useful. Broader impacts: It is important that the authors properly address the possible positive and negative societal impacts of their work, as opposed to simply re-stating the main contributions of the paper. Minor (not affecting my score): * Line 120: I am not sure what “expert system” means in this context. Does this term simply imply that \xi^d is optimal? * Equation 4: What is the expectation over? * There are a number of typos in the paper, which should be fixed.


Review 4

Summary and Contributions: The paper proposes Pontryagin differentiable programming (PDP) as a learning and control framework to address a variety learning and control problems, including inverse reinforcement learning (IRL) and inverse control (IOC).

Strengths: + The paper provides an insightful discussion on the relationship of learning and control. + The proposed approach is novel and theoretically sound. The formulated problem is general that can be applied to various application domains. + Theoretical analysis is comprehensive. + Experimental results are convincing. + The paper is well organized and well written.

Weaknesses: - If we care about the internal control law of a system and remove the assumption J(\theta) = 0 in lines 127-128, whether does the proposed approach work? - It is not clear why Eq (4) aims to/is able to learn a reward function in inverse reinforcement learning. - Please define end-to-end and explain why the proposed approach is end-to-end? The term end-to-end is defined and most widely used in deep learning, e.g., to accomplish representation learning and classification using one deep network. The meaning of end-to-end is not clear in this paper. - Please define and explain explainable in line 91. Explainable is used in ML as the opposite side of a deep network that is considered as a black box. Since this approach is not DL, then comparing with other conventional RL or control methods, why the proposed approach is more explainable? - Similarly, the meaning of structural in line 92 is also not defined, which seems to have a different meaning than structural learning in ML. - In experiments, demonstrations for imitation learning are provided by a control with optimal parameters. This raises a concern about how the approach can be used in practice when demonstrations are far less structured with significant variations (e.g., from human demonstrators). Please explain.

Correctness: Theoretical analysis of the approach is sound.

Clarity: The paper is well organized and well written.

Relation to Prior Work: Difference from previous work is clear.

Reproducibility: Yes

Additional Feedback:


Review 5

Summary and Contributions: This paper presents a framework of end-to-end learning for control systems. This work introduces the differentiable system model and proposes a method based on Pontryagin’s Maximum principle. The benefits of the proposed framework are demonstrated in the context of system identification, imitation learning and optimal control.

Strengths: The proposed method introduces the structure into the neural network in a principled way, and the methodology is theoretically sound. The proposed way of incorporating inductive bias may inspire the audiences in the NeurIPS community.

Weaknesses: - One of the limitations of the proposed method is about the assumption. Although the methodology is technically sound, the proposed method requires the prior knowledge on the system dynamics; The form of the differential equations that governs the environment needs to be known, although the parameters may be unknown. I think that the assumption that the form of the differential equations are known is very strong and it may be impractical. For example, it is hard to describe the behavior of deformable objects. Meanwhile, I'm also aware that this is a common limitation in this line of work. - A weak point of this work is the empirical evaluation. Although the proposed method is compared with some baseline methods, compared methods are not recent ones. GAIL may be a more suitable baseline than inverse KKT in the imitation learning experiment, and guided policy search can be also better baseline than simple iLQR in the control experiment.

Correctness: Comparison with existing methods that incorporate differentiable equations with a neural network. For example, the following papers are missing: [1]S. Sæmundsson, A. Terenin, K. Hofmann, M. P. Deisenroth (2020). Variational Integrator Networks for Physically Structured Embeddings. Proceedings of the International Conference on Artificial Intelligence and Statistics (AISTATS). [2]M. Raissi, P. Perdikaris, and G. E. Karniadakis. Physicsinformed neural networks: a deep learning framework for solving forward and inverse problems involving nonlinear partial differential equations. Journal of Computational Physics, 378(686–707), 2019 [3]Lutter, M.; Ritter, C.; Peters, J. (2019). Deep Lagrangian Networks: Using Physics as Model Prior for Deep Learning, International Conference on Learning Representations (ICLR) As a result, it is hard to assess the benefit of the proposed method compared to existing method that incorporates the differential equations to a neural network. Although the paper addresses wide range of problems including IRL, OC, and system identification, some of recent studies are missing in the related work section. In the context of imitation learning, Generative Adversarial Imitation Learning (GAIL) and its variants have been investigated by recent studies. However, comparison with such existing methods are not compared in the experiments. [4] J. Ho and S. Ermon. (2016) Generative Adversarial Imitation Learning, NeurIPS 2016.

Clarity: Although the paper is easy to follow, it would be hard to reproduce the results due to the lack of detailed information. For example, the overall network structure is not clear.

Relation to Prior Work: As described in the above box, the discussion on the relation to the existing methods are missing. Especially, the relation to [1-3] should be discussed in the related work section.

Reproducibility: No

Additional Feedback: I recommend the authors to describe the neural network structure and the training conditions. Otherwise, it is hard to reproduce the reported results. === comments after rebuttal === I have read the other reviews and author response. I appreciate the authors' efforts for adding more baselines to the experiments. I think that the claims in the paper is supported by the experiments firmly. In addition, the motivation of this study is clearly explained in the rebuttal. I expect the authors to improve the background and the motivation in the paper. I raise the score to 7.

[Author Response · NeurIPS 2020]

**Response to Rev. #1: (A)** We have performed comparison with other end-to-end learning frameworks in Appendix F, Page 11, including comparison with 'differentiable MPC'. In this response, we also provide additional comparison with 'differentiable MPC' in Fig. 1. In revision, we will move all comparisons to the primary text. **(B)** We try to unify the fundamental IRL, OC, SysID problems, as they can be viewed the same problem with different locations of unknown parameters. Background currently reviews the most related control/learning methods for these problems. In revision, we will discuss other related topics like Bayesian models, MDP, and MPC. **(C)** 'Control/Planning mode' formulates a typical OC problem, which can be solved by iLQR, while PDP is *a new OC method* and empirically shows more efficiency than iLQR. MPC is rather a strategy of solving OC with shifting horizons, and its implementation still requires OC solvers. **(E)** the neural policy is to imitate a policy, while inverse KKT learns a cost function. With limited data (see Appendix E), the former can be over-fitting and the latter generally has *better generality to unseen conditions*, as a cost function is a high-level representation of policies.

**Response to Rev. #2: (A)** PDP allows the use of (deep) neural networks to represent unknowns of a system. As supportive examples, we use neural networks to re-do experiments of IRL, SysID, and OC in Fig. 1 (will be added to revision). **(B)** PDP uses gradient descent to solve non-convex bi-level optimization, and only local optima are achieved. But if we make assumptions, e.g., convexity/smoothness, on inner OC and outer loss, we could have global convergence by bi-level programming theory [S. Ghadimi, et al, Approximation Methods for Bilevel Programming. arXiv:1802.02246]. However, such conditions are too restrictive to inner OC. In the future, we will investigate mild conditions using control (e.g., Lyapunov) theory. **(C)** Empirically, parameterization matters in its convergence. E.g., neural network introduces high non-linearity and is more likely to trap in local optima than polynomial; thus simply parameterization is preferred. **(D)** A failure source of PDP is catastrophic error of OC solver – causing error of trajectory gradient. Please see Appendix H for other discussions. **(E)** All codes will be released.

**Response to Rev. #3: (A)** We will mathematically declare Equ (2) as argmin of (1) in place of language exposition (here we followed exposition conventions in control literature). **(B)** The reviewer makes an interesting comment about the needs of having intermediate Jacobian $\frac{\partial \xi_\theta}{\partial \theta}$. Before explaining why, we argue that PDP is also capable of directly computing $\frac{\partial l}{\partial \theta}$ without needing $\frac{\partial \xi_\theta}{\partial \theta}$. Consider Equ (13) (stack for all $t$) as a big linear equation for $\frac{\partial x_{0:T}^\theta}{\partial \theta}$, $\frac{\partial u_{0:T}^\theta}{\partial \theta}$, and $\frac{\partial \lambda_{1:T}^\theta}{\partial \theta}$. By multiplying the inverse of coefficient matrix of this equation to the backward pass vector $\frac{\partial l}{\partial \xi}$ from loss, we obtain the analytical gradient $\frac{\partial l}{\partial \theta} = \frac{\partial l}{\partial \xi}\frac{\partial \xi_\theta}{\partial \theta}$ without explicit $\frac{\partial \xi_\theta}{\partial \theta}$, similar to Equ (7) in OptNet (Equ (9) in differentiable MPC) and Neural ODE. However, we have not adopted this way, because it *causes huge computation and memory complexity*: the size of this linear equation (Equ (13)) is $(m+n)T \times (m+n)T$ ($T$ is the time horizon, usually very large, $m, n$ are state, input dims, also see Equ (6) in differentiable MPC); and computing the inverse of such a big matrix costs at least a complexity of $(m+n)^2 T^2$ (e.g., in our UAV experiment, it requires to solve a 1000*1000-sized linear equation). Due to these inefficiency, PDP proposes to explicitly solve the intermediate Jacobian $\frac{\partial \xi}{\partial \theta}$ using auxiliary control system, where the memory is only $2(m+n)T$ and computation is also $2(m+n)T$ thanks to recursion structure. Aside benefit: a side-product of solving $\frac{\partial \xi_\theta}{\partial \theta}$ is that *gradient of individual trajectory point* $\frac{\partial x_t^\theta}{\partial \theta}$ is obtained. It enables to *learn from sparse data* as shown in Fig. 1e : in IRL, given sparse demo $x_t^d$ only at time $t$, PDP can use $\frac{\partial x_t^\theta}{\partial \theta}$ to $\min_\theta \|x_t^\theta - x_t^d\|^2$. **(C)** In Appendix F, Page 11, we have stated the above advantage and reported the timing of PDP in Fig. 7 compared to OptNet (differentiable MPC): *PDP is much faster than OptNet*. **(D)** The revision will cite all mentioned papers, improve Sections 2, 3, and address the minor issues. **(E)** For reproducibility, please find experiment/algorithm details in Appendix D/E. Codes will be released.

**Response to Rev. #5: (A)** If both *dynamics* and *feedback policy* are parameterized, PDP still applies. But we shall be cautious here: if we set the loss as imitation loss as in Equ (6), it is fine but more desirable to learn each separately by supervised learning; if we define the loss as a control cost like Equ (8), very likely we will arrive at trivial (zero) models since parameterizing both is redundant. **(B)** Equ (4) states: $\min_\theta \|\xi_\theta - \xi^d\|$ s.t. $\xi_\theta$ is the optimal trajectory of OC system $\Sigma(\theta)$ in Equ (1). It says that we want to find a cost function $J(\theta)$ in the OC system such that its produced optimal trajectory $\xi_\theta$ is closest to the demonstration $\xi^d$. **(C)** The end-to-end concept in our context refers to a way to look at PDP framework as a whole neural network (representing a control system). Then PDP trains this neural network by directly minimizing the loss over the parameter of interest using gradient descent. However, unlike generic NNs, we inject the inductive bias to this PDP NN using optimal control theory: the trajectory of OC satisfies PMP, which makes the PDP NN more efficient to solve control/learning problems. **(D)** Explainability is relative to the classic RL frameworks, where the key is to learn a less-interpretable value function. Differently, PDP attacks control/learning problems from trajectory optimization perspective (PMP), bypassing the need to solve value functions. Also, the trajectory has a direct meaning of system behavior. **(E)** Structure means that PDP investigates the relation of *states/inputs at level of each time step*, versus existing black-box NNs, which treat the entire trajectory as a single variable (see Appendix F, Page 11). **(F)** Given demonstration of significant variations, PDP can still find the 'best' objective function within the parameterized function set $J(\theta)$ such that its produced $\xi_\theta$ has *minimal distance* to the demonstrations. Please see a supportive experiment in Fig. 1e.

**Response to Rev. #6: (A)** PDP does not limit the parameterization of dynamics and allows it to have any differentiable parameterization. Given little physical knowledge of system dynamics, one can represent it by generic neural networks (but would be less efficient to learn than physics-based parameterization), as in Fig. 1c. **(B)** As additional comparison, we here compare PDP with the suggested GAIL and guided policy search. Please see Fig. 1 (will be added to revision). The results show that PDP outperforms GAIL and guided policy search for higher efficiency, lower training loss, and better generality. **(C)** The suggested three references are about how to inject physical laws into construction of physics-informed NNs for learning physically-plausible ODEs. Differently, PDP is not concerned about how to obtain physical-inspired NNs; instead, PDP considers a *generic mathematical parameterization of a control system* and how to efficiently train such parameterized control system. We inject the inductive bias of optimal control theory to PDP framework: the trajectory of OC satisfies PMP, which makes the PDP suitable to efficiently solve control/learning problems. **(D)** The parameterization in PDP can be, of course, the physics-informed NNs e.g. deep Lagrangian networks. Although in submission, the experiments use parameterized physical dynamics, PDP allows for using neural networks to implement dynamics/policy/control-objective as in Fig. 1. **(E)** For reproducibility/clarity, we kindly refer the reviewer to Appendix D, Page 3, for algorithm/implementation details, Appendix E, Page 4, for experiment details, Appendix F, Page 11, for relationship with other learning frameworks. All the mentioned references will be discussed in revision.

Figure 1: Support experiment (robot arm): parameterized control-objective/dynamics/policy are implemented by fully-connected neural networks with 2 hidden layers with 20 nodes each. PDP outperforms others. Dataset/other parameters follow Appendix E in supplementary.

[Meta-Review · NeurIPS 2020]

This paper proposes a unified framework for solving inverse RL, system identification, and optimal control problems, using implicit differentiation through Pontryagin’s Maximum Principle (PMP). The paper applies this framework to imitation learning, system identification, and optimal control tasks in four experimental domains, and shows improved performance compared to other methods. According to R1, who has a more traditional control theory background, in their effort to present this unified view of these problems, the authors make the implicit claim that the method is equally useful in all three modes: system identification, optimal control, and inverse optimal control. In reality, the proposed method presents an inductive bias from the PMP that is most useful for the first and third modes (SysID and IOC), where the current dominant approach is the inverse KKT method from Toussaint and others and bespoke SysID methods. In terms of optimal control, the proposed unification seems more of an abstraction and interpretation, but does not offer a new method. That said, it is necessary to include it in the paper because it is a building block for the other modes. Reviewers said that this paper is novel and exciting enough to be accepted, and I definitely agree. I am not aware of other methods that differentiate through the PMP.